# Automatic Visual Instrumental Variable Learning for Confounding-Resistant Domain Generalization

**Fuyuan Cao[1, 2], Shichang Qiao[1],\* Kui Yu[3], Jiye Liang[1]**
[1]School of Computer and Information Technology, Key Laboratory of Computational
Intelligence and Chinese Information Processing of Ministry of Education,
Shanxi University, Taiyuan, China
[2]Shanxi Taihang Laboratory, Taiyuan, China
[3]School of Computer Science and Information Engineering,
Hefei University of Technology, Hefei, China
`cfy@sxu.edu.cn, qiaoshichang@sxu.edu.cn,`
`yukui@hfut.edu.cn, ljy@sxu.edu.cn`

## Abstract

Many confounding-resistant domain generalization methods for image classification have been developed based on causal interventions. However, their reliance on strong assumptions limits their effectiveness in handling unobserved confounders. Although recent work introduces instrumental variables (IVs) to overcome this limitation, the reliance on manually predefined instruments, particularly in the context of visual data, may result in severe bias or invalidity when IV conditions are violated. To address these issues, we propose a novel approach to automatically learning **V**isual **I**nstrumental **V**ariables for confounding-resistant **D**omain **G**eneralization (VIV-DG). We observe that certain non-causal visual attributes in image data naturally satisfy the basic conditions required for valid IVs. Motivated by this insight, we propose the *visual instrumental variable*, a novel concept that extends classical IV theory to the visual domain. Furthermore, we develop an automatic visual instrumental variable learner that enforces IV conditions on learned representations, enabling the automatic learning of valid visual instrumental variables from image data. Ultimately, VIV-DG inherits the strengths of classical IVs to mitigate unobserved confounding and avoids the significant bias caused by violations of IV conditions in predefined IVs. Extensive experiments on multiple benchmarks verify that VIV-DG achieves superior generalization ability.

## 1 Introduction

In machine learning, enhancing model generalization under distribution shifts remains a critical challenge[1, 2]. To tackle this issue, domain generalization (DG) has emerged as a prominent research area that aims to extract knowledge from source domains and effectively generalize to unseen target domains [3, 4]. Recently, numerous DG methods have been proposed, such as adversarial learning [5], domain augmentation [6, 7], invariant representation learning [8], explicit feature alignment [9], and meta-learning [10, 11, 12], which have achieved significant progress. However, these methods primarily rely on statistical correlations, which may still be insufficient for addressing domain shifts [13]. This is because domain shifts are typically accompanied by confounders that simultaneously influence both features and labels, introducing confounding effects and creating spurious correlations between them. Consequently, methods based purely on statistical correlations may inadvertently model these spurious associations, compromising their generalization capability.

---

\*Corresponding author.

39th Conference on Neural Information Processing Systems (NeurIPS 2025).

To avoid spurious correlations, researchers employ causal intervention techniques [14] (such as backdoor adjustment, frontdoor adjustment, and instrumental variable methods) to mitigate confounding effects and learn cross-domain invariant causal representations. However, these techniques rely on strong assumptions and face significant limitations in real-world applications. For example, DIR-ReID [15] adopts backdoor adjustment to develop a more generalizable person re-identification framework. Yet, backdoor adjustment assumes that all confounders are observable, while unobserved confounders are commonly present in practice, limiting its effectiveness. FAGT [16] utilizes front-door adjustment to mitigate confounding effects, but depends on valid mediators, which are often difficult to identify and validate in many tasks. In contrast, instrumental variables (IVs) provide an alternative solution that avoids explicitly modeling confounders or relying on mediators. IV-DG [17] incorporates the instrumental variable framework into domain generalization by adopting a predefined strategy that treats data from one domain as instruments for another. However, such predefined instruments often violate the exclusion restriction. Although they originate from different domains, features extracted by pretrained encoders for images of the same category typically exhibit cross-domain discriminability. This violates the core assumption of IVs, rendering them weak or invalid, and ultimately impairing generalization performance. In addition, predefined IVs rely heavily on expert knowledge and require substantial human effort. Taken together, existing methods remain inadequate in mitigating confounding effects, especially those caused by unobserved confounders.

Moreover, as shown in Figure 1a and Figure 1b, existing DG methods based on eliminating confounding effects often rely on prior structural causal models (SCMs) to partition inputs into causal factors (such as shape and contour) and non-causal factors (such as background, style, and color), treating all non-causal factors as confounders to be suppressed. However, this coarse-grained division and the paradigm of

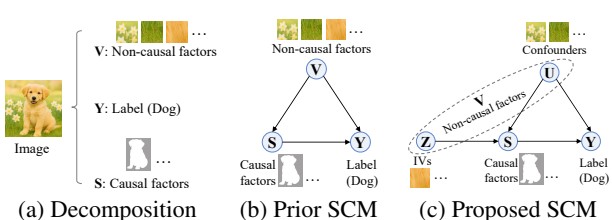

(a) Decomposition    (b) Prior SCM    (c) Proposed SCM

Figure 1: Comparison of proposed and prior SCM.

uniformly treating all non-causal factors as confounders have notable limitations. In image data, there often exist non-causal features that are stable across domains (e.g., the color and texture of the main object). If these features are indiscriminately treated as confounders along with domain-specific factors (such as background color), the model's discriminative capability may be undermined. For instance, although object color and texture may not be sufficient on their own to determine category labels (e.g., yellow patches from a yellow cat and a yellow dog may look similar), they can significantly enhance discrimination when combined with shape and contour cues.

In this paper, we propose a novel approach, VIV-DG (**V**isual **I**nstrumental **V**ariable for **D**omain **G**eneralization), which effectively mitigates confounding effects and preserves valuable non-causal information by automatically learning visual instrumental variables from image data without relying on strong assumptions. However, obtaining valid IVs remains a significant challenge. Fortunately, we observe that image data typically contain some non-causal visual attributes that satisfy the conditions for IVs. Inspired by these observations, we unify the two critical challenges—identifying valid instrumental variables and preserving discriminative non-causal features—into a single learning objective: autonomously learning representations from images that satisfy the IV conditions and using IV regression to mitigate confounding effects. Specifically, as depicted in Figure 1c, we finely categorize non-causal factors into confounders and IVs, and establish a more refined SCM to analyze the causal relationships among variables. Furthermore, we construct a learning framework for IVs, composed of three alternately optimized subnetworks: (i) a causal feature extractor that extracts causal representations and performs classification tasks; (ii) a visual instrumental variable learner that learns valid IV representations by enforcing relevance, independence, and exclusion constraints; (iii) a regression predictor that predicts causal factors unaffected by confounding effects. The entire framework is trained with an alternating optimization strategy, progressively yielding more refined IV and causal representations. Ultimately, VIV-DG inherits the advantages of classical IVs in mitigating unobserved confounding, while avoiding the significant bias that may arise when predefined IVs violate their conditions. Moreover, it reduces excessive reliance on expert knowledge and the high cost of manual design. Experiments on multiple benchmarks show that VIV-DG significantly enhances domain generalization performance. To sum up, the contributions of our work are as follows:

- We propose VIV-DG, a novel approach that automatically learns visual instrumental variables to effectively mitigate the effects of both observed and unobserved confounders, resulting in improved domain generalization.

- We define the novel concept of visual instrumental variables and develop a learner that automatically learns valid ones, effectively overcoming the severe bias caused by violations of IV conditions in predefined approaches.

- Extensive experiments on multiple real-world benchmarks verify the effectiveness and advantages of VIV-DG, demonstrating improved generalization ability.

## 2 Motivation

We draw inspiration from two essential observations, which collectively serve as the foundation for the development of our approach.

**Intrinsic differences among non-causal factors.** We observe that non-causal factors in image data exhibit substantial intrinsic differences. As shown in Figure 2, local textures and color patches of an object, although not directly determining the category, can provide discriminative features when combined with causal factors such as shape and contour, aligning with human cognitive perception. In contrast, combining background elements like grass or flowers with object contours often leads to biased or inconsistent recognition.

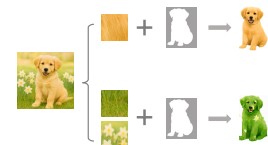

Figure 2: Non-causal factors comparison.

**Certain visual attributes serving as IVs.** We further find that when causal factors such as object shape and contour are treated as treatment variables, intrinsic visual attributes of the object, such as color and texture, often satisfy the conditions for IVs [18]: (i) *Relevance*: Object color and texture are biologically or physically associated with shape and contour; (ii) *Independence*: Object color and texture are primarily determined by inherent genetic or physical properties, and are thus independent of confounders; (iii) *Exclusion*: Object color and texture do not directly determine the category label but influence classification indirectly by affecting the object's shape and contour.

## 3 Methodology

We propose the VIV-DG approach and develop a multi-stage alternating optimization strategy to progressively refine both visual IVs and causal representations. The framework illustrated in Figure 3.

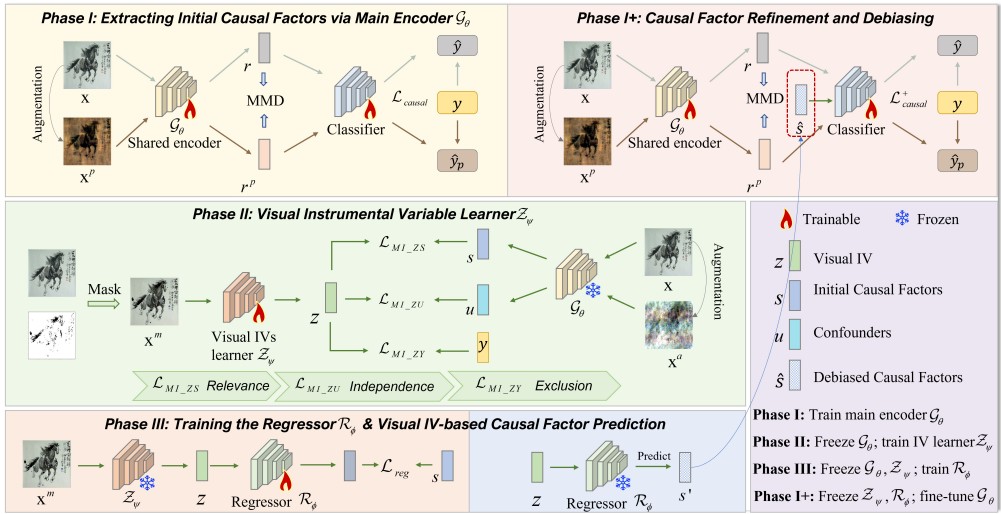

Figure 3: Overview of the VIV-DG framework, which consists of multiple alternately optimized phases: (I) Initial causal factor extraction, (II) Visual IV learning under IV constraints, (III) Causal prediction robust to confounding, and (I+) Causal factor refinement and debiasing.

## 3.1 Visual instrumental variable

Observing that certain visual attributes naturally satisfy instrumental variable conditions, we propose the concept of visual instrumental variable (Visual IV) to better exploit such valuable information.

**Definition 1** (Visual instrumental variable). *Suppose the visual space contains a triplet $(X, Y, S)$, where $X$ denotes a visual object (e.g., an image), $Y$ is the label for a downstream task, and $S$ represents a specific causal factor in the visual object such that $S \rightarrow Y$. A visual attribute $Z$ is defined as visual instrumental variable (Visual IV) if it satisfies the following three conditions:*

*(i) Relevance: $Z$ is significantly associated with the causal factor $S$, i.e., $Z \not\perp\!\!\!\perp S$;*

*(ii) Independence: $Z$ is independent of the confounders $U$, i.e., $Z \perp\!\!\!\perp U$;*

*(iii) Exclusion: $Z$ affects $Y$ only through its influence on the causal factor $S$, i.e., $Z \perp\!\!\!\perp Y \mid S$.*

## 3.2 Proposed SCM for DG with IVs

Building on the Visual IVs, we establish a refined SCM that explicitly incorporates both IVs and unobserved confounders, as shown in Figure 4. In this model, $X$ represents the input data (e.g., images), and $Y$ represents the class labels, while $U$, $C$, $Z$, and $S$ collectively constitute the generative factors of the input data. Specifically, $S$ represents causal factors (e.g., the shape and contours of the object), while $U$, $C$, and $Z$ are non-causal factors. Here, $U$ denotes observed confounders (e.g., the background and style of the current image); $C$ represents unobserved confounders (e.g., visual factors that vary across

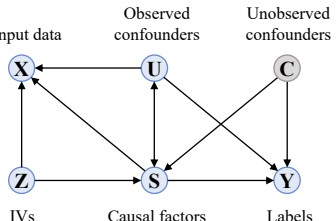

Figure 4: Proposed SCM for DG.

unseen domains); and $Z$ denotes Visual IVs (e.g., object color and texture) that influence $X$ but do not directly affect $Y$. All the detailed connections in the SCM are presented in the Appendix C.

Notably, since the confounders $U$ and $C$ simultaneously affect both the causal factors $S$ and the labels $Y$, they establish backdoor paths between $S$ and $Y$: $S \leftrightarrow U \rightarrow Y$ and $S \leftarrow C \rightarrow Y$. As a result, in addition to the causal relationship $S \rightarrow Y$, spurious correlations are introduced, leading to confounding effects [19, 20]. Conventional estimation of $P(Y|S)$ results in a biased estimator that absorbs both the true causal dependence and spurious correlations, limiting its generalizability.

## 3.3 Disentanglement of causal factors and confounders

Guided by the proposed SCM, we employ the widely used Fourier-based data augmentation to learn representations of the initial causal factors (mainly including the shape and contour information of the object) and confounders.

**Causal factor extraction.** The phase component obtained via the Fourier transform [21] of an image primarily encodes semantic (i.e., discriminative) information, whereas the amplitude component mainly captures style-related (i.e., non-discriminative) details [13]. Based on this observation, we perturb the amplitude of the original image $\mathbf{x}_i$ while preserving its phase, and generate an augmented version denoted as $\mathbf{x}_i^p$. We then learn the consistent representation of $\mathbf{x}_i$ and $\mathbf{x}_i^p$ as the initial causal factor through a consistency constraint using maximum mean discrepancy (MMD) [22]. Specifically, for an image $\mathbf{x}_i$, its Fourier transform $\mathcal{F}(\mathbf{x}_i)$ is represented as

$$\mathcal{F}(\mathbf{x}_i) = \mathcal{A}(\mathbf{x}_i) \times e^{-j \times \mathcal{P}(\mathbf{x}_i)}, \tag{1}$$

where $\mathcal{A}(\mathbf{x}_i)$ and $\mathcal{P}(\mathbf{x}_i)$ denote the amplitude and phase components, respectively. To generate the augmented image $\mathbf{x}_i^p$ that emphasizes phase information, we perturb the amplitude component as

$$\mathcal{A}'(\mathbf{x}_i) = (1 - \lambda_1)\mathcal{A}(\mathbf{x}_i) + \lambda_1 \mathcal{A}(\hat{\mathbf{x}}_j), \tag{2}$$

where $\hat{\mathbf{x}}_j$ is randomly sampled from an arbitrary source domain, and $\lambda_1 \sim U(0.5, 1)$ controls the perturbation strength. We then reconstruct the image by combining the perturbed amplitude $\mathcal{A}'(\mathbf{x}_i)$ with the original phase $\mathcal{P}(\mathbf{x}_i)$ through inverse Fourier transform:

$$\mathbf{x}_i^p = \mathcal{F}^{-1}\left(\mathcal{A}'(\mathbf{x}_i) \times e^{-j \times \mathcal{P}(\mathbf{x}_i)}\right). \tag{3}$$

To learn consistent representation (i.e., the initial causal factor) between the images $\mathbf{x}_i$ and $\mathbf{x}_i^p$, we employ a shared encoder $\mathcal{G}_\theta(\cdot)$ to extract their features: $r_i = \mathcal{G}_\theta(\mathbf{x}_i)$ and $r_i^p = \mathcal{G}_\theta(\mathbf{x}_i^p)$. We then leverage the MMD [22] to enforce representation consistency. The MMD loss is defined as

$$\mathcal{L}_{\text{MMD}} = \left\| \frac{1}{n_1} \sum_{i=1}^{n_1} \Phi\left(\mathbf{x}_i\right) - \frac{1}{n_2} \sum_{i=1}^{n_2} \Phi\left(\mathbf{x}_i^p\right) \right\|_{\mathcal{H}}^2, \tag{4}$$

where $n_1$ and $n_2$ denote the numbers of original and augmented samples, respectively. Here, $n_1 = n_2$. The function $\Phi(\cdot)$ maps inputs to a reproducing kernel Hilbert space (RKHS), denoted by $\mathcal{H}$.

To capture class-discriminative information from both the original image $\mathbf{x}$ and the augmented image $\mathbf{x}^p$, we apply a classifier $h$ to the extracted features $r$ and $r^p$, producing predicted label distributions $\hat{y}$. The classification loss is computed using the standard cross-entropy:

$$\mathcal{L}_{\text{cls}} = \mathbb{E}_{(\mathbf{x},y) \sim p(\mathbf{x},y)} \left[ -y \log \sigma_m(f(\mathbf{x};\theta)) \right] + \mathbb{E}_{(\mathbf{x}^p,y^p) \sim p(\mathbf{x}^p,y^p)} \left[ -y^p \log \sigma_m\left(f\left(\mathbf{x}^p;\theta\right)\right) \right], \tag{5}$$

where $\sigma_m$ is the softmax activation and $f(\cdot)$ denotes $h \circ g(\cdot)$. To further encourage representation consistency, we incorporate the $\mathcal{L}_{\text{MMD}}$ loss, and define the overall loss for learning causal factors as

$$\mathcal{L}_{\text{causal}} = \beta_1 \mathcal{L}_{\text{cls}} + \tau \mathcal{L}_{\text{MMD}}, \tag{6}$$

where $\beta_1$ and $\tau$ are trade-off hyperparameters. The learned consistent representation is used as the initial causal representation, i.e., the initial causal factor $s_i$, for the original image $\mathbf{x}_i$.

**Confounder extraction.** To extract confounders without relying on domain labels, we also adopt the Fourier-based data augmentation strategy. In contrast to causal factors, confounders typically capture domain-specific characteristics such as color and style, which are primarily reflected in the amplitude spectrum. Thus, we perturb the phase information and combine it with the original image's amplitude $\mathcal{A}(\mathbf{x}_i)$ to generate the augmented image $\mathbf{x}_i^a$ that retains the amplitude information:

$$\mathcal{P}'\left(\mathbf{x}_i\right) = (1 - \lambda_2)\mathcal{P}\left(\mathbf{x}_i\right) + \lambda_2 \mathcal{P}\left(\hat{\mathbf{x}}_j\right), \tag{7}$$

$$\mathbf{x}_i^a = \mathcal{F}^{-1}\left(\mathcal{A}\left(\mathbf{x}_i\right) \times e^{-j \times \mathcal{P}'(\mathbf{x}_i)}\right), \tag{8}$$

where $\mathcal{P}'(\mathbf{x}_i)$ denotes the perturbed phase components, and $\lambda_2 \sim U(0.5, 1)$ controls the perturbation strength. We then derive the confounder representation $u_i$ by extracting features from the augmented image $\mathbf{x}_i^a$ through the encoder $\mathcal{G}_\theta(\cdot)$, i.e., $u_i := r_i^a = \mathcal{G}_\theta\left(\mathbf{x}_i^a\right)$.

**Remark 1.** *Unlike the correlation matrix loss used in CIRL [13], which enforces both consistency between original and augmented causal features and decorrelation across different samples, we adopt the MMD loss to focus solely on the former. We observe that enforcing inter-sample independence may discard shared causal cues. For instance, zebras and horses share similar shapes, which are generally considered causal features [23]. In this case, emphasizing feature independence may weaken the representation of shape-related causal information.*

## 3.4 Automatic Visual IV learning

### 3.4.1 Masked image construction for IV representation learning

To ensure that the learned Visual IV representations compensate for the texture and color features while excluding shape and contour information, we design a mask generation module to preprocess the input data. This module consists of two main components: (i) *image difference computation and mask generation*, and (ii) *masked image construction*. Specifically, this module preserves texture and color-related regions by computing pixel-wise differences between grayscale versions of the original and augmented images. It then masks out regions dominated by shape information. This targeted preprocessing procedure ensures that the learned Visual IV representations capture attributes complementary to causal features. We denote the masked image as $\mathbf{x}^m$, and extract its representation using the IV encoder $\mathcal{Z}_\psi$, denoted by $z := \mathcal{Z}_\psi(\mathbf{x}^m)$. The details are provided in Appendix D.

### 3.4.2 Relevance-constrained learning

We enforce that Visual IV $Z$ remains highly informative of the causal factors $S$ by maximizing their mutual information $I(Z;S)$. Mutual information is formally defined as [24]

$$I(Z;S) = \mathbb{E}_{p(z,s)} \left[ \log \frac{p(z,s)}{p(z)p(s)} \right], \tag{9}$$

which is intractable in high dimensions because it requires the true joint and marginal densities. Drawing inspiration from classifier-based variational bounds [25], we employ an adversarial training scheme that leverages positive and negative sample pairs: Positive pairs $\{(z_i, s_i)\}_{i=1}^N$ are drawn from the same original sample. Negative pairs $\{(z_i, s_j)\}_{i \neq j}$ are from different image samples. Furthermore, we construct a discriminator network $T_\mu : \mathbb{R}^{2d} \to \mathbb{R}$ and apply a sigmoid activation $\sigma_s(\cdot) \in (0, 1)$ to interpret the output as the probability of a positive pair. The mutual information admits the following lower bound based on the Jensen–Shannon divergence variational principle:

$$I(Z; S) \geq \mathbb{E}_{p(z,s)}\left[-\log\left(1 + e^{-T_\mu(z,s)}\right)\right] + \mathbb{E}_{p(z)p(s)}\left[-\log\left(1 + e^{T_\mu(z,s)}\right)\right]. \tag{10}$$

We operationalize this bound through a minibatch-approximated binary cross-entropy objective:

$$\mathcal{L}_{\mathrm{MI\_ZS}} = -\frac{1}{N}\sum_{i=1}^N \log \sigma_s\left(T_\mu\left(z_i, s_i\right)\right) - \frac{1}{N}\sum_{i=1}^N \frac{1}{N-1}\sum_{j \neq i} \log\left(1 - \sigma_s\left(T_\mu\left(z_i, s_j\right)\right)\right). \tag{11}$$

The first term encourages $\sigma_s(T_\mu(z_i, s_i))$ to approach 1 for positive pairs, while the second term drives $\sigma_s(T_\mu(z_i, s_j))$ toward 0 for negative pairs. For notational clarity, we further express the objective in its expectation form:

$$\mathcal{L}_{\mathrm{MI\_ZS}} = -\mathbb{E}_{(z,s)\sim p(z,s)}[\log \sigma_s\left(T_\mu(z, s)\right)] - \mathbb{E}_{(z,s)\sim p(z)p(s)}[\log\left(1 - \sigma_s\left(T_\mu(z, s)\right)\right)]. \tag{12}$$

By minimizing $\mathcal{L}_{\mathrm{MI\_ZS}}$, we effectively maximize the variational lower bound on $I(Z; S)$, thereby satisfying the relevance condition of the Visual IV.

### 3.4.3 Independence-constrained learning

To enforce statistical independence between the learned Visual IV $Z$ and the confounders $U$, we aim to minimize their mutual information $I(Z; U)$. Direct computation of $I(Z; U)$ is intractable in high-dimensional space, so we adopt an adversarial approach based on the Jensen–Shannon (JS) divergence. Specifically, we introduce a discriminator network $T'_\mu : \mathbb{R}^{2d} \to \mathbb{R}$ that distinguishes samples from the joint distribution $p(z, u)$ and the product of marginals $p(z)p(u)$. Positive pairs $\{(z_i, u_i)\}_{i=1}^N$ are drawn from $p(z, u)$, preserving the original correlation, while negative pairs $\{(z_i, u_j)\}_{i \neq j}$ are generated by randomly shuffling the confounder indices. To confuse the discriminator and reduce mutual information, we define the empirical adversarial loss as

$$\mathcal{L}_{\mathrm{MI\_ZU}} = \frac{1}{N}\sum_{i=1}^N \log \sigma_s(T'_\mu(z_i, u_i)) + \frac{1}{N}\sum_{i=1}^N \frac{1}{N-1}\sum_{j \neq i} \log(1 - \sigma_s(T'_\mu(z_i, u_j))), \tag{13}$$

where $\sigma_s(\cdot)$ denotes the sigmoid function. The equivalent expectation form is

$$\mathcal{L}_{\mathrm{MI\_ZU}} = \mathbb{E}_{(z,u)\sim p(z,u)}[\log \sigma_s(T'_\mu(z, u))] + \mathbb{E}_{(z,u)\sim p(z)p(u)}[\log(1 - \sigma_s(T'_\mu(z, u)))]. \tag{14}$$

Minimizing this loss with respect to the Visual IV encoder confuses the discriminator, thereby reducing $I(Z; U)$ and enforcing the independence condition for the Visual IV.

### 3.4.4 Exclusion-constrained learning

To ensure the learned Visual IV $Z$ satisfies the exclusion condition (i.e., $Z$ affects the outcome $Y$ solely through the causal factors $S$), we propose a variational approximation to minimize the conditional mutual information $I(Y; Z \mid S)$. Since direct computation in high-dimensional spaces is intractable, we approximate it by minimizing the Kullback-Leibler (KL) divergence between two conditional distributions. Formally, the conditional mutual information is defined as

$$I(Y; Z \mid S) = \mathbb{E}_{p(s,z)}\left[D_{\mathrm{KL}}(p(Y \mid s, z) \,\|\, p(Y \mid s))\right] = H(Y \mid S) - H(Y \mid S, Z), \tag{15}$$

where $H(\cdot)$ denotes conditional entropy and $D_{\mathrm{KL}}(\cdot\|\cdot)$ denotes KL divergence. If $Z$ remains dependent on $Y$ given $S$, then $H(Y \mid S, Z) < H(Y \mid S)$, implying $I(Y; Z \mid S) > 0$. To enforce that $Z$ carries no predictive information beyond $S$, we minimize this divergence term. In practice, we use the main classifier $h$ to produce differentiable proxies for the true conditional distributions:

$$p_s(y|s) = \sigma_m(h(s)), \quad p_{z\oplus s}(y|z, s) = \sigma_m(h(z \oplus s)), \tag{16}$$

where $\oplus$ denotes element-wise addition (adopted instead of concatenation to avoid dimensionality explosion), and $\sigma_m$ represents the softmax function. If the causal factor $S$ sufficiently encodes information relevant to predicting $Y$, the ideal scenario satisfies

$$p_{z\oplus s}(y|z,s) = p_s(y|s) \quad \forall z, \tag{17}$$

which yields $I(Y; Z \mid S) = 0$. Thus, we enforce this constraint by minimizing the KL divergence between the two distributions:

$$\mathcal{L}_{\text{MI\_ZY}} = \mathbb{E}_{p(s,z)} D_{\text{KL}}\left(p_{z\oplus s} \| p_s\right) = \mathbb{E}\left[\sum_y p_{z\oplus s}(y \mid z, s) \log \frac{p_{z\oplus s}(y \mid z, s)}{p_s(y \mid s)}\right]. \tag{18}$$

Minimizing $\mathcal{L}_{\text{MI\_ZY}}$ is equivalent to maximizing $H(Y|S,Z)$, which encourages $Z$ to preserve the uncertainty of $Y$ given $S$. This mechanism enforces the exclusion restriction by guaranteeing that the instrument $Z$ can only affect the outcome $Y$ through the causal pathway mediated by $S$.

### 3.4.5 Overall objective function for Visual IV learning

The overall objective for Visual IV representation learning is thus formulated as

$$\mathcal{L}_{\text{total\_IV}} = \alpha_1 \mathcal{L}_{\text{MI\_ZS}} + \alpha_2 \mathcal{L}_{\text{MI\_ZU}} + \alpha_3 \mathcal{L}_{\text{MI\_ZY}}, \tag{19}$$

where $\alpha_1$, $\alpha_2$, and $\alpha_3$ are trade-off hyperparameters. During backpropagation, only the parameters of the Visual IV learner $\mathcal{Z}_\psi$ and the discriminator are updated, while the parameters of both the main encoder $\mathcal{G}_\theta$ and the classifier $h$ are kept frozen. Notably, although derived from the images, the learned IVs can still exhibit exogeneity through disentangled learning.

### 3.5 Regression predictor for causal factor refinement and debiasing

We construct a lightweight regression predictor $\mathcal{R}_\phi$ (abbreviated as the regressor) that maps the learned Visual IV representation $z$ back to the original causal feature space, predicting new causal factors (causal representations) that are unaffected by confounding effects.

Regressor $\mathcal{R}_\phi$ is implemented as a three-layer MLP with LayerNorm, BatchNorm, GELU activation, and Dropout. Given the layer-normalized input $z$, It outputs the estimated reconstructed causal factors, $\hat{s} := \mathcal{R}_\phi(\text{LN}(z))$, where $\text{LN}(\cdot)$ denotes layer normalization. We train $\mathcal{R}_\phi$ by minimizing the mean-squared error between $\hat{s}$ and the learned initial causal factors $s$. The loss is expressed as

$$\mathcal{L}_{\text{reg}} = \mathbb{E}_{(s,z)\sim p(s,z)}\left[\|\mathcal{R}_\phi(\text{LN}(z)) - s\|_2^2\right]. \tag{20}$$

During this stage, both the main encoder $\mathcal{G}_\theta$ and the Visual IV learner $\mathcal{Z}_\psi$ remain frozen; only the parameters of regressor $\mathcal{R}_\phi$ are updated. After training the regressor $\mathcal{R}_\phi$, we freeze the parameters of both the IV learner $\mathcal{Z}_\psi$ and the regressor $\mathcal{R}_\phi$. We then input the debiased causal factor $\hat{s}$, predicted by $\mathcal{R}_\phi$, into the main classifier $h$ and compute the corresponding classification loss:

$$\mathcal{L}_{\text{cls}}^{\hat{s}} = \mathbb{E}_{(\hat{s},y)\sim p(\hat{s},y)}\left[-y \log \sigma_m h(\hat{s})\right], \tag{21}$$

where $\sigma_m$ denotes the softmax function. Subsequently, we unfreeze the main encoder $\mathcal{G}_\theta$ and classifier $h$, and jointly optimize the classification loss $\mathcal{L}_{\text{cls}}$ on the original image $\mathbf{x}$ and its phase-augmented images $\mathbf{x}^p$, along with the MMD loss $\mathcal{L}_{\text{MMD}}$, to obtain the refined causal representation learning loss:

$$\mathcal{L}_{\text{causal}}^+ = \beta_1 \mathcal{L}_{\text{cls}} + \beta_2 \mathcal{L}_{\text{cls}}^{\hat{s}} + \tau \mathcal{L}_{\text{MMD}}. \tag{22}$$

By optimizing the above objective, we correct and debias the main encoder $\mathcal{G}_\theta$, encouraging the model to learn more comprehensive causal representations and thus improve generalization.

### 3.6 Progressive optimization via four-stage alternating training

We adopt a four-stage training protocol to optimize each component: (i) we first train the primary encoder $\mathcal{G}_\theta$ to extract initial causal factors; (ii) we then freeze $\mathcal{G}_\theta$ and train the IV learner $\mathcal{Z}_\psi$ to learn Visual IVs; (iii) next, we freeze both $\mathcal{G}_\theta$ and $\mathcal{Z}_\psi$, and train the regressor $\mathcal{R}_\phi$ to establish a mapping from Visual IVs to causal factors; (iv) finally, we freeze $\mathcal{Z}_\psi$ and $\mathcal{R}_\phi$, and fine-tune $\mathcal{G}_\theta$ to further refine causal representation learning. In particular, the third and fourth stages reflect the causal logic of IV regression. Notably, although the model components are optimized in alternating stages, the overall framework is end-to-end trainable. For clarity, the stage-wise configuration is summarized in Appendix H.3.4. The pseudocode for VIV-DG is presented in Appendix F.

**Remark 2** (Limitation–Solution). *To address the limitation that training the regression predictor increases computational cost when using larger backbones for the main encoder $\mathcal{G}_\theta$ and the Visual IV learner $\mathcal{Z}_\psi$, we propose a simplified alternative: a feature fusion strategy that integrates the learned Visual IV features and the initial causal factors via addition in the representation space. We refer to this lightweight version as **VIV-DG-Lite**. Notably, the fusion reduces training cost but compromises interpretability, whereas the regressor offers better interpretability at the expense of model complexity.*

## 4 Theoretical analysis

We present the main theoretical guarantee, with formal proofs provided in Appendix E.

**Theorem 1** (Learnability of Visual IVs). *Let $(X, S, U, Y)$ be four random variables, where $X$ denotes observed images, $S$ denotes the causal factors affecting $Y$, $U$ denotes the confounders, and $Y$ denotes the downstream labels. Assume that $\mathcal{H}_Z = \{h_\omega : X \to Z\}$ is a sufficiently expressive family of mappings (e.g., one that contains all smooth bijections on a latent subspace). Consider the objective*

$$\mathcal{L}(\omega) = -\alpha_1 I(h_\omega(X); S) + \alpha_2 I(h_\omega(X); U) + \alpha_3 I(h_\omega(X); Y \mid S), \tag{23}$$

*with $\alpha_1, \alpha_2, \alpha_3 > 0$. Then any global minimizer $\omega^*$ yields*

$$Z^* = h_{\omega^*}(X), \tag{24}$$

*which is learnable as defined in the learnability definition (Definition 2) in the Appendix.*

## 5 Experiments

### 5.1 Datasets and settings

We evaluate VIV-DG on several real-world benchmarks: Digits-DG [26], PACS [27], Office-Home [28], and VLCS [29]. The datasets and implementation details are shown in Appendix H.

### 5.2 Experimental results

#### 5.2.1 Evaluation on benchmarks

We compare non-causal and causality-inspired DG methods. A detailed description of the baselines is provided in Appendix H.2. Table 1 presents the results of our VIV-DG and its simplified version, VIV-DG-Lite, on the Digits-DG, PACS (ResNet-18), and Office-Home (ResNet-18) datasets.

Table 1: Leave-one-domain-out accuracies (%) on Digits-DG, PACS, and Office-Home

| Methods | Digits-DG | | | | | PACS (ResNet-18) | | | | | Office-Home (ResNet-18) | | | | |
|---|---|---|---|---|---|---|---|---|---|---|---|---|---|---|---|
| | MN | MM | SVHN | SYN | Avg. | A | C | P | S | Avg. | A | C | P | R | Avg. |
| DeepAll [26] | 95.8 | 58.8 | 61.7 | 78.6 | 73.7 | 77.6 | 76.8 | 95.9 | 69.5 | 79.9 | 57.9 | 52.7 | 73.5 | 74.8 | 64.7 |
| CCSA [30] | 95.2 | 58.2 | 65.5 | 79.1 | 74.5 | - | - | - | - | - | 59.9 | 49.9 | 74.1 | 75.7 | 64.9 |
| JiGen [31] | 96.5 | 61.4 | 63.7 | 74.0 | 73.9 | 79.4 | 75.3 | 96.0 | 71.4 | 80.5 | 53.0 | 47.5 | 71.5 | 72.8 | 61.2 |
| RSC [32] | - | - | - | - | - | 83.4 | 80.3 | 96.0 | 80.9 | 85.2 | 58.4 | 47.9 | 71.6 | 74.5 | 63.1 |
| CrossGrad [33] | 96.7 | 61.1 | 65.3 | 80.2 | 75.8 | - | - | - | - | - | 58.4 | 49.4 | 73.9 | 75.8 | 64.4 |
| DDAIG [26] | 96.6 | 64.1 | 68.6 | 81.0 | 77.6 | 84.2 | 78.1 | 95.3 | 74.7 | 83.1 | 59.2 | 52.3 | 74.6 | 76.0 | 65.5 |
| MatchDG [34] | - | - | - | - | - | 81.3 | 80.7 | 96.5 | 79.7 | 84.6 | - | - | - | - | - |
| L2A-OT [35] | 96.7 | 63.9 | 68.6 | 83.2 | 78.1 | 83.3 | 78.2 | 96.2 | 73.6 | 82.8 | 60.6 | 50.1 | 74.8 | **77.0** | 65.6 |
| CIRL [13] | 96.1 | **69.9** | 76.2 | 87.7 | 82.5 | 86.1 | 80.6 | 95.9 | 82.7 | 86.3 | 61.5 | 55.3 | 75.1 | 76.6 | 67.1 |
| LRDG [36] | - | - | - | - | - | 81.9 | 80.2 | 95.2 | **84.7** | 85.5 | **61.7** | 52.4 | 73.0 | 75.9 | 65.8 |
| FACT [7] | 97.9 | 65.6 | 72.4 | 90.3 | 81.5 | 85.9 | 79.4 | 96.6 | 80.9 | 85.7 | 60.3 | 54.9 | 74.5 | 76.6 | 66.6 |
| IV-DG [17] | - | - | - | - | - | 83.4 | 78.8 | **96.9** | 78.7 | 84.4 | 60.4 | 47.7 | 72.6 | 76.1 | 64.2 |
| FAGT [16] | 98.3 | 65.7 | 70.9 | 90.4 | 81.3 | **87.5** | 80.9 | **96.9** | 81.9 | 86.8 | 60.1 | 55.0 | 74.5 | 75.8 | 66.4 |
| CDIM [37] | **98.7** | 64.0 | 74.1 | **92.9** | 82.4 | 83.6 | 77.6 | 95.5 | 78.2 | 83.7 | - | - | - | - | - |
| VIV-DG-Lite | 97.8 | 66.8 | **77.8** | 91.4 | **83.5** | 86.1 | **81.8** | 96.6 | 83.3 | 87.0 | 60.8 | 55.9 | **75.8** | 76.3 | 67.2 |
| VIV-DG | 97.6 | 67.1 | 77.4 | 91.6 | 83.4 | 86.6 | 81.5 | **96.9** | 83.9 | **87.2** | 61.1 | **56.3** | 75.3 | 76.8 | **67.4** |

† The best and second best results are marked in bold and underlined, respectively. Avg. = Average accuracy(%).

Table 1 shows that our method exhibits two key advantages: (i) it outperforms existing causality-inspired DG methods, with significant improvements over the IV-DG [17] method, which relies on the predefined IV strategy; (ii) it demonstrates a pronounced advantage under severe distribution

shifts. For instance, on the Digits-DG dataset, where *SVHN* and *SYN* exhibit large distribution gaps, VIV-DG and VIV-DG-Lite outperform the baselines by a clear margin. Similarly, in PACS, where there is a substantial distribution shift, our method consistently surpasses all baselines.

For a more comprehensive validation, we evaluate our method using a larger backbone network, ResNet-50, to assess its scalability and robustness. Table 2 presents the results on the VLCS dataset. It can be observed that both VIV-DG and VIV-DG-Lite achieve the same optimal average performance, surpassing recent causality-inspired methods with gains of 0.8% over iDAG and 0.5% over GMDG. Moreover, they demonstrate competitive performance across multiple domains. Table 1 and Table 2 demonstrate that our methods effectively mitigate confounding effects and improve generalization.

## 5.3 Analysis of the distinct behaviors of VIV-DG and VIV-DG-Lite

As shown in Table 1 and Table 2, the performance of VIV-DG and VIV-DG-Lite varies across target domains. To understand this discrepancy, we provide a detailed analysis. Specifically, VIV-DG-Lite approximates the causal representations by combining the IVs with the initial causal representations, which retains more original discriminative information but is more susceptible to confounding. Therefore, it performs better in domains with smaller distribution shifts, such as *Product* in Office-Home and *Cartoon* in PACS dataset. In contrast, VIV-DG predicts causal representations through a trained

Table 2: The results on VLCS with ResNet-50

| Methods | C | L | S | P | Avg. |
|---|---|---|---|---|---|
| RSC [32] | 98.0 | 67.2 | 70.3 | 75.6 | 77.8 |
| MixStyle [38] | 98.6 | 64.5 | 72.6 | 75.7 | 77.9 |
| SagNet [39] | 97.9 | 64.5 | 71.4 | 77.5 | 77.8 |
| PCL [40] | 99.0 | 63.6 | 73.8 | 75.6 | 78.0 |
| iDAG [41] | 98.1 | 62.7 | 69.9 | 77.1 | 76.9 |
| RICE [42] | 98.3 | **69.2** | 74.6 | 75.1 | 79.3 |
| GMDG [43] | 98.3 | 65.9 | 73.4 | **79.3** | 79.2 |
| VIV-DG-Lite | **99.2** | 67.4 | 74.4 | 77.6 | **79.7** |
| VIV-DG | 99.0 | 67.7 | **74.6** | 77.5 | **79.7** |

regressor for bias correction, which helps reduce confounding effects and thus performs better in domains with larger distribution shifts, such as *Sketch* in PACS dataset. Overall, VIV-DG demonstrates greater robustness compared to VIV-DG-Lite.

## 5.4 Analytical experiments

### 5.4.1 Parameter sensitivity

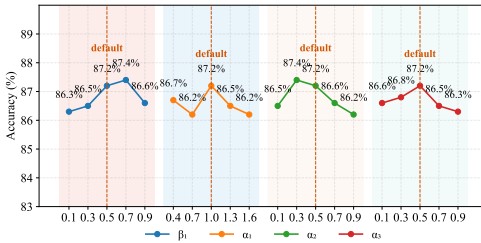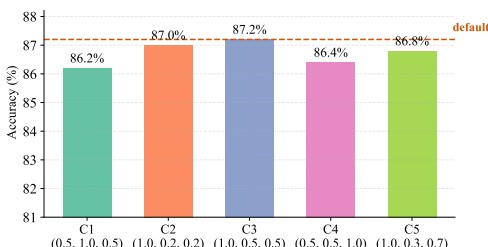

Figure 5: (Left) Univariate hyperparameter analysis. (Right) Combined hyperparameter analysis.

To evaluate the robustness of the VIV-DG under different hyperparameter settings, we perform a sensitivity analysis on the PACS dataset using the ResNet-18. Appendix H.3.3 summarizes the hyperparameter settings for each stage. Since $\tau$ and $\beta_2$ are adaptively adjusted, the analysis focuses on $\beta_1$, $\alpha_1$, $\alpha_2$, and $\alpha_3$. We first perform a univariate sensitivity analysis for each hyperparameter, followed by a combination analysis of $\alpha_1$, $\alpha_2$, and $\alpha_3$, keeping other hyperparameters at their default values. The value ranges and step sizes are listed in Appendix H.4. As shown in Figure 5, the model demonstrates satisfactory robustness across both individual and combined hyperparameter settings.

### 5.4.2 Ablation study

To evaluate the contribution of each component, we conduct ablation studies on Digits-DG, PACS, and Office-Home. As shown in Table 3, removing both the Visual IV module and the regressor (w/o VIV & $\mathcal{R}_\phi$) leads to a significant performance drop compared to the full model, indicating that relying solely on data augmentation and MMD for invariant representation learning is insufficient. This demonstrates that mitigating confounding effects is crucial for effective domain generalization. Moreover, the results show that the exclusion loss is the most critical, as it effectively prevents the

introduction of extra confounders as IVs. Meanwhile, the relevance and independence losses help reduce confounding effects and enhance the expressiveness of causal representations.

Table 3: Ablation study (%) on Digits-DG, PACS, and Office-Home datasets

| Methods | Setting | Digits-DG | | | | | PACS (ResNet-18) | | | | | Office-Home (ResNet-18) | | | | |
|---|---|---|---|---|---|---|---|---|---|---|---|---|---|---|---|---|
| | | MN | MM | SV | SY | Avg. | A | C | P | S | Avg. | A | C | P | R | Avg. |
| VIV-DG | w/o VIV & $\mathcal{R}_\phi$ | 96.7 | 62.3 | 72.8 | 89.7 | 80.4 | 82.7 | 78.1 | 93.6 | 80.5 | 83.7 | 58.7 | 54.6 | 74.2 | 76.1 | 65.9 |
| | w/o $I(Z;S)$ | 96.8 | 64.5 | 74.0 | 90.6 | 81.5 | 85.5 | 80.4 | 95.8 | 83.6 | 86.3 | 60.9 | 54.4 | 75.6 | 76.7 | 66.9 |
| | w/o $I(Z;U)$ | 96.9 | 64.1 | 75.0 | 90.2 | 81.6 | 84.9 | 79.7 | 95.9 | 83.0 | 85.9 | 60.9 | 53.9 | 75.3 | 76.6 | 66.7 |
| | w/o $I(Y;Z|S)$ | 96.7 | 64.6 | 74.7 | 90.5 | 81.6 | 84.6 | 79.1 | 95.3 | 82.4 | 85.4 | 60.7 | 53.6 | 75.4 | 76.5 | 66.6 |
| | Full Model | 97.6 | 67.1 | 77.4 | 91.6 | 83.4 | 86.6 | 81.5 | 96.9 | 83.9 | 87.2 | 61.1 | 56.3 | 75.3 | 76.8 | 67.4 |
| VIV-DG-Lite | w/o VIV & $\mathcal{R}_\phi$ | 96.7 | 62.3 | 72.8 | 89.7 | 80.4 | 82.7 | 78.1 | 93.6 | 80.5 | 83.7 | 58.7 | 54.6 | 74.2 | 76.1 | 65.9 |
| | w/o $I(Z;S)$ | 97.1 | 64.4 | 74.9 | 90.4 | 81.7 | 85.6 | 79.7 | 95.4 | 82.9 | 85.9 | 60.5 | 53.9 | 75.4 | 75.9 | 66.4 |
| | w/o $I(Z;U)$ | 96.9 | 63.9 | 74.6 | 90.2 | 81.4 | 84.7 | 78.4 | 95.5 | 82.4 | 85.3 | 60.4 | 53.7 | 75.6 | 76.1 | 66.5 |
| | w/o $I(Y;Z|S)$ | 96.5 | 64.2 | 74.4 | 90.1 | 81.3 | 84.5 | 78.2 | 95.5 | 81.9 | 85.0 | 60.2 | 53.3 | 75.2 | 75.7 | 66.1 |
| | Full Model | 97.8 | 66.8 | 77.8 | 91.4 | 83.5 | 86.1 | 81.8 | 96.6 | 83.3 | 87.0 | 60.8 | 55.9 | 75.8 | 76.3 | 67.2 |

† The results of "w/o IV & $\mathcal{R}_\phi$" are identical for both VIV-DG and VIV-DG-Lite under this setting, and are provided for both methods to facilitate direct comparison.

### 5.4.3 Visual explanation

To validate that VIV-DG learns causal representations consistent with human cognition, we conduct visual analysis using Grad-CAM on images with complex backgrounds from the *Art-Painting* domain. We deliberately choose this domain because the classification accuracies of VIV-DG and the baseline CIRL [13] are comparable, making any visual improvement in identifying causal regions a meaningful contribution. We compare VIV-DG against both CIRL and an ablated version of our model without the Visual IV learning module (Ours (w/o VIV)). As shown in Figure 6, VIV-DG demonstrates a superior ability to capture global features and focus on the core object structures. For instance, in images of *dog*, *giraffe*, and *house*, VIV-DG identifies broader, more holistic regions compared to the localized cues relied upon by CIRL. More importantly, the ablated model (Ours (w/o VIV)) exhibits significant deviations, especially in complex backgrounds (e.g., *dog*). Even in simpler scenes (e.g., *horse* and *house*), it is less accurate than the full model in capturing causally relevant regions. These results provide strong visual evidence that the Visual IV module is crucial, further supporting VIV-DG's effectiveness in learning cognitively consistent causal representations.

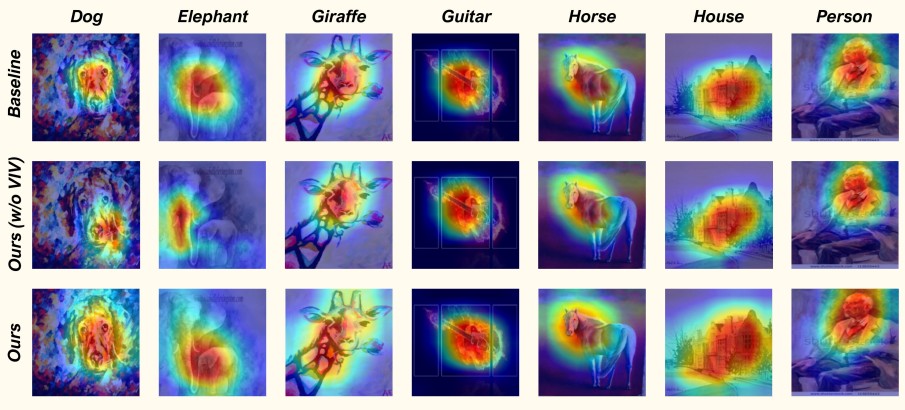

Figure 6: Grad-CAM visualization on the PACS dataset with *Art-Painting* as the target domain.

## 6 Conclusion

We propose a novel confounding-resistant DG approach, termed VIV-DG. We break away from the conventional view that simply categorizes non-causal factors as confounders and observe that certain visual attributes in image data satisfy the conditions of IVs. Building on this insight, we develop a framework that automatically learns valid Visual IVs and mitigates the significant bias arising from violations of IV conditions in predefined IVs. By mitigating confounding effects, including those from unobserved confounders, VIV-DG consistently achieves improved generalization.

## Acknowledgments

This work was supported by the National Science and Technology Major Project of China (2021ZD0111801), the National Natural Science Foundation of China (U24A20323, 62376145, and 62376087), the Open Project Special Fund of Taihang Laboratory In Shanxi Province, China (THYF-KFKT-25020200), the Key Technologies Program of Taihang Laboratory in Shanxi Province (THYF-JSZX-24010700), the Science and Technology Innovation Talent Team of Shanxi Province (202204051002016), and the Taiyuan City "Double hundred Research action" of the first batch project about "Leading the Charge with Open Competition" (2024TYJB0127).

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

# Appendix

This appendix includes technical and supplementary materials: related work, technical background, the VIV-DG pseudocode, experimental setup, additional results, and broader impacts.

## A    Related work

### A.1    Confounding effect mitigation for DG

Recently, numerous domain generalization methods have been developed to learn causal representations by mitigating confounding effects. These methods are typically categorized into three causal intervention frameworks: backdoor adjustment, frontdoor adjustment, and instrumental variable (IV) methods. For instance, Zhang et al. [15] propose the DIR-ReID framework based on backdoor adjustment to learn domain-invariant representations for person re-identification. Similarly, Sui et al. [44] design a causal attention mechanism using backdoor adjustment to improve robustness and interpretability. However, backdoor adjustment assumes access to all confounders, which is often unrealistic due to the presence of unobserved confounders in real-world scenarios. Toan Nguyen et al. [16] propose the FAGT method based on frontdoor adjustment, achieving improved generalization. Nonetheless, it relies on observable mediators, which are difficult to identify and validate in practice. Yuan et al. [17] introduce IV-DG, which treats data from one domain as instrumental variables for another. However, even if domains differ, features extracted from the same class may still share cross-domain discriminative information, potentially violating the exclusivity assumption of IVs.

### A.2    Instrumental variable

Instrumental variables are exogenous variables that are associated with the treatment but have no direct causal effect on the outcome [18]. Even in the presence of unobserved confounders, IV methods can effectively eliminate confounding effects, which makes instrumental variables widely applicable for causal effect estimation and counterfactual prediction [45, 46]. Recently, instrumental variables have gained increasing attention in machine learning. In addition to the previously mentioned domain generalization method IV-DG[17], instrumental variables are also applied in recommendation systems. For example, Si et al. [47] propose the IV4Rec method, which leverages user search data as instrumental variables to effectively mitigate confounding bias caused by latent variables in recommendation models.

In our work, we introduce a domain generalization approach for image classification that directly learns instrumental variables from image data. Our method is not restricted by specific physical concepts but instead autonomously learns representations that satisfy instrumental variable conditions. The key advantage of our approach is that predefined instrumental variables may introduce severe biases if they fail to meet valid IV conditions, whereas our method effectively avoids this issue by autonomously learning instrumental variables.

## B    Preliminary: instrumental variables for confounding mitigation

Instrumental Variables (IVs) [18] are a statistical method that can effectively estimate causal effects even in the presence of unobserved confounders. The core idea is to introduce a exogenous variable that satisfies specific conditions, thereby blocking the confounding effect on the relationship between the treatment $T$ and the outcome $Y$. To illustrate the properties and role of IVs, we present a general SCM, as shown in Figure 7. A valid instrumental variable must satisfy three conditions: (i) Relevance: The variable $Z$ must be correlated with the treatment $T$; (ii) Independence: The variable $Z$ must be independent of the confounder $C$; (iii) Exclusion: The variable $Z$ affects the outcome $Y$ only through the treatment $T$ (i.e., $Z$ has no direct effect on $Y$).

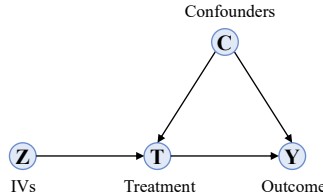

Figure 7: The SCM with IVs.

The most widely used IV-based causal effect estimation method is two-stage least squares regression (2SLS), which follows a two-stage procedure. In the first stage, the treatment $T$ is regressed on the instrument $Z$, yielding the predicted treatment values $\hat{T}$. In the second stage, the outcome Y

is regressed on the predicted values $\hat{T}$. The coefficient obtained in the second-stage regression represents the local average treatment effect (LATE), providing an unbiased estimate of the causal effect of the $T$ on $Y$. Notably, 2SLS is primarily designed for scenarios with linear relationships between variables.

## C Connections of the proposed SCM

Figure 8 illustrates our proposed fine-grained SCM, with all the connections detailed below.

$S \rightarrow X$, $U \rightarrow X$, $Z \rightarrow X$. $(S, U, Z) \rightarrow X$. Each input image $X$ consists of causal factors (i.e., object-specific factors) $S$ and non-causal factors $U$ and $Z$. Although both $U$ and $Z$ are non-causal factors, they differ fundamentally. As an inherent attribute of the object, $Z$ remains relatively stable and, when combined with $S$, provides valuable information for category prediction.

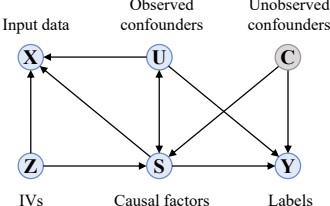

Figure 8: The proposed SCM.

$S \leftrightarrow U \rightarrow Y$, $S \leftarrow C \rightarrow Y$. The observed confounders $U$ should not be directly associated with $Y$. However, in practice, when selection bias exists in the training data, pretrained encoders may learn shortcut features from non-causal factors and use them for prediction. As a result, $U$ influences $Y$, even though this relationship is non-causal. Additionally, due to natural or manual biases, $S$ and $U$ frequently co-occur in the data distribution, leading to a statistical association between them. We represent this correlation with a bidirectional arrow $(S \leftrightarrow U)$. Moreover, $C$ shares similarities with $U$ but represents unobserved confounders that simultaneously influence both $S$ and $Y$. Without loss of generality, we assume that $C$ represents confounders related to unknown domain styles or backgrounds. Since domains cannot be exhaustively enumerated, such unobserved confounders inevitably exist. However, these factors do not appear in the current observed input data or known source domains, and therefore do not exert a direct causal effect on the input data $X$. For example, factors such as image style or background color may serve as potential confounders, but they are not always visible or relevant in specific samples. The art style of an image, as a confounder, may not affect the input of a sketch image. Similarly, a background factor like yellow desert does not appear in an image of a dog in the snow.

$Z \rightarrow S \rightarrow Y$. The factors $Z$ and $S$ may exhibit a biological or physical association. For example, in animal images, this could reflect the co-evolution of an organism's coat color and shape. Additionally, $Z$ does not directly influence the labels. For instance, it is not possible to determine the category of an object solely based on its local coat color. However, when combined with shape and contours, $Z$ provides valuable predictive information.

Notably, since the confounders $U$ and $C$ simultaneously affect both the causal factors $S$ and the labels $Y$, they establish backdoor paths between $S$ and $Y$: $S \leftrightarrow U \rightarrow Y$ and $S \leftarrow C \rightarrow Y$. As a result, in addition to the causal relationship $S \rightarrow Y$, spurious correlations are introduced, leading to confounding effects [19]. Conventional estimation of $P(Y|S)$ results in a biased estimator that absorbs both the true causal dependence and spurious correlations, limiting its generalizability.

## D Mask generation module

The mask generation module consists of two main components: (i) image difference computation and mask generation, and (ii) masked image construction.

(i) Image difference computation and mask generation: We convert both the original image $\mathbf{x}_i$ and its augmented counterpart $\mathbf{x}_i^p$ to grayscale to reduce the influence of color. Then, we compute the absolute pixel-wise difference $\Delta_i(\mathrm{p}, \mathrm{q})$ at each location $(\mathrm{p}, \mathrm{q})$:

$$\Delta_i(\mathrm{p}, \mathrm{q}) = \left| \mathbf{x}_{i(g)}(\mathrm{p}, \mathrm{q}) - \mathbf{x}_{i(g)}^p(\mathrm{p}, \mathrm{q}) \right|, \tag{25}$$

where $\mathbf{x}_{i(g)}$ and $\mathbf{x}_{i(g)}^p$ represent the transformed grayscale images, respectively. A binary mask is then generated using a threshold $\delta$ (empirically set to 30):

$$m_i(\mathrm{p}, \mathrm{q}) = \begin{cases} 1, & \Delta_i(\mathrm{p}, \mathrm{q}) < \delta, \\ 0, & \Delta_i(\mathrm{p}, \mathrm{q}) \geq \delta, \end{cases} \quad \delta = 30. \tag{26}$$

(ii) Masked image construction: We extend the binary mask $m_i(\mathrm{p}, \mathrm{q})$ to all RGB channels and apply it to the original image $\mathbf{x}_i$ to obtain the masked input:

$$\mathbf{x}_i^m(\mathrm{p}, \mathrm{q}, \mathrm{c}) = \mathbf{x}_i(\mathrm{p}, \mathrm{q}, \mathrm{c}) \times m_i(\mathrm{p}, \mathrm{q}), \tag{27}$$

where c denotes the color channel. The masked image $\mathbf{x}_i^m$ is then fed into the IV learner to extract the IV representations.

# E   Theoretical foundations

## E.1   Preliminaries and assumptions

We consider a tuple of random variables $(X, S, U, Y)$ defined on a probability space $(\Omega, \mathcal{F}, \mathbb{P})$, where $X$ denotes the observed images, $S$ denotes the causal factors affecting $Y$, $U$ denotes the confounders, and $Y$ denotes the downstream labels. Let $\mathcal{Z}$ be a measurable representation space, and let $Z = h(X)$ for some representation function $h : X \to \mathcal{Z}$. Without loss of generality, we do not distinguish between observed and unobserved confounders here.

**Assumption 1** (Basic Regularity). *The joint laws of all relevant random variables admit regular conditional distributions, and all mutual informations appearing below are finite. Moreover, all measurable maps and densities are taken modulo null sets (i.e., hold almost surely).*

**Assumption 2** (Parametric Representation Family; Continuity and Compactness). *The representation family is parameterized as $\mathcal{H}_Z = \{h_\omega : X \to \mathcal{Z}\}_{\omega \in \Omega}$ where (i) the parameter set $\Omega$ is compact, (ii) the map $\omega \mapsto h_\omega$ is measurable, and (iii) for each of the mutual-information functionals*

$$\omega \mapsto I(h_\omega(X); S), \quad \omega \mapsto I(h_\omega(X); U), \quad \omega \mapsto I(h_\omega(X); Y \mid S), \tag{28}$$

*we assume continuity (or at least the required lower/upper semicontinuity) so that objectives built from these functionals admit minimizers on $\Omega$ by standard compactness arguments.*

## E.2   Identifiability conditions and learnability

**lemma 1** (Identifiability Conditions for Visual IVs via Mutual Information). *Let $(X, S, U, Y)$ be four random variables, where $X$ denotes observed images, $S$ denotes the causal factors affecting $Y$, $U$ denotes the confounders, and $Y$ denotes the downstream labels. A candidate variable $Z$ is called a visual instrumental variable if it simultaneously satisfies: (i) Relevance: $I(Z; S) > 0$, (ii) Independence: $I(Z; U) = 0$, and (iii) Exclusion: $I(Z; Y \mid S) = 0$.*

*Proof.* By definition of mutual information,

$$I(Z; S) = \iint p(z, s) \log \frac{p(z, s)}{p(z)p(s)} \, dz \, ds, \tag{29}$$

and $I(Z; S) > 0$ if and only if $p(z, s) \neq p(z)p(s)$, i.e., $Z \not\perp\!\!\!\perp S$. This is exactly the relevance requirement. Similarly,

$$I(Z; U) = \iint p(z, u) \log \frac{p(z, u)}{p(z)p(u)} \, dz \, du, \tag{30}$$

and $I(Z; U) = 0$ if and only if $p(z, u) = p(z)p(u)$, i.e., $Z \perp\!\!\!\perp U$. This is the independence requirement. Finally,

$$I(Z; Y \mid S) = \iiint p(s, y, z) \log \frac{p(y, z \mid s)}{p(y \mid s)p(z \mid s)} \, ds \, dy \, dz, \tag{31}$$

and $I(Z; Y \mid S) = 0$ if and only if $p(y, z \mid s) = p(y \mid s)p(z \mid s)$, i.e., $Z \perp\!\!\!\perp Y \mid S$. This is the exclusion requirement. $\square$

**lemma 2** (Recoverability of Visual IVs). *Let $h_\omega : X \to Z$ be a parameterized mapping (e.g., via a deep network), with the training objective*

$$\mathcal{L}(\omega) = -\alpha_1 I(Z; S) + \alpha_2 I(Z; U) + \alpha_3 I(Z; Y \mid S), \tag{32}$$

*where $\alpha_1, \alpha_2, \alpha_3 > 0$. Under the idealized assumption of sufficient network capacity, any global minimizer $\omega^*$ satisfies*

$$I(Z^*; S) > 0, \quad I(Z^*; U) = 0, \quad I(Z^*; Y \mid S) = 0, \tag{33}$$

*where $Z^* = h_{\omega^*}(X)$, i.e., it exactly recovers a visual instrumental variable.*

*Proof.* Since $-I(Z;S) \geq -\sup_\omega I(Z;S)$, $I(Z;U) \geq 0$, and $I(Z;Y \mid S) \geq 0$, any global minimizer $\omega^*$ must simultaneously (i) maximize $I(Z;S)$, (ii) drive $I(Z;U)$ to zero, and (iii) drive $I(Z;Y \mid S)$ to zero; otherwise one could reduce $\mathcal{L}$ further.

By the zero mutual information conditions, we get $Z \perp\!\!\!\perp U$ and $Z \perp\!\!\!\perp Y \mid S$, and by the maximization, $I(Z;S) > 0$. These three facts are exactly the three requirements of Lemma 1. $\square$

### E.3 Identifiability theory

**Definition 2** (Learnability of Visual IVs)**.** *Let $(X, S, U, Y)$ be four random variables, where $X$ denotes observed images, $S$ denotes the causal factors affecting $Y$, $U$ denotes the confounders, and $Y$ denotes the downstream labels. Let $\mathcal{H}_Z = \{h : X \to Z\}$ be a family of representation functions. We say that a visual instrumental variable is learnable in $\mathcal{H}_Z$ if:*

*(1) (Existence) There exists at least one $h^* \in \mathcal{H}_Z$ whose output $Z^* = h^*(X)$ satisfies the three conditions of Lemma 1:*

$$I(Z^*; S) > 0, \quad I(Z^*; U) = 0, \quad I(Z^*; Y \mid S) = 0. \tag{34}$$

*(2) (Uniqueness modulo invertible transforms) For any two $h_1, h_2 \in \mathcal{H}_Z$ whose outputs both satisfy those three conditions and maximize $I(Z;S)$ (i.e., $Z$ is a sufficient statistic for $S$), their outputs are related by an invertible reparameterization:*

$$h_2(X) = \varphi(h_1(X)), \quad \varphi : \mathcal{Z} \to \mathcal{Z} \text{ bijective}. \tag{35}$$

**Proposition 1** (Uniqueness of Visual IVs)**.** *Let $\mathcal{H}_Z = \{h : X \to Z\}$ be a sufficiently rich function family (containing all smooth bijections). Suppose some $h^* \in \mathcal{H}_Z$ satisfies*

$$I(Z^*; S) > 0, \quad I(Z^*; U) = 0, \quad I(Z^*; Y \mid S) = 0, \tag{36}$$

*where $Z^* = h^*(X)$, and $I(Z^*; S)$ is maximized (i.e., $Z^*$ is a sufficient statistic for $S$). Then the induced variable $Z^*$ is unique up to invertible reparameterization.*

*Proof.* Let $Z_1 = h_1(X)$ and $Z_2 = h_2(X)$ both satisfy the conditions. Since $I(Z_1; U) = 0$ and $I(Z_1; Y \mid S) = 0$, $Z_1$ is independent of $U$ and conditionally independent of $Y$ given $S$. Therefore, $Z_1$ can only influence $Y$ through $S$. Similarly, $Z_2$ can only influence $Y$ through $S$. Since $I(Z_1; S)$ and $I(Z_2; S)$ are both maximized, $Z_1$ and $Z_2$ are both sufficient statistics for $S$. This means there exist functions $g_1$ and $g_2$ such that $S = g_1(Z_1)$ and $S = g_2(Z_2)$ almost surely (in an appropriate sense). Consequently, $Z_1$ and $Z_2$ determine each other through $S$: there exist mappings $f_{12}$ and $f_{21}$ satisfying $Z_2 = f_{12}(Z_1) = f_{12}(g_1^{-1}(S))$ and $Z_1 = f_{21}(Z_2) = f_{21}(g_2^{-1}(S))$ Since the function family $\mathcal{H}_Z$ is sufficiently rich (containing all smooth bijections), $f_{12}$ and $f_{21}$ can be taken as bijective transformations. Thus, $Z_1$ and $Z_2$ are equivalent up to an invertible reparameterization. $\square$

**Theorem 2** (Learnability of Visual IVs)**.** *Let $(X, S, U, Y)$ be four random variables, where $X$ denotes observed images, $S$ denotes the causal factors affecting $Y$, $U$ denotes the confounders, and $Y$ denotes the downstream labels. Assume that $\mathcal{H}_Z = \{h_\omega : X \to Z\}$ is a sufficiently expressive family of mappings (e.g., one that contains all smooth bijections on a latent subspace). Consider the objective*

$$\mathcal{L}(\omega) = -\alpha_1 I(h_\omega(X); S) + \alpha_2 I(h_\omega(X); U) + \alpha_3 I(h_\omega(X); Y \mid S), \tag{37}$$

*with $\alpha_1, \alpha_2, \alpha_3 > 0$. Then any global minimizer $\omega^*$ yields*

$$Z^* = h_{\omega^*}(X), \tag{38}$$

*which is learnable as defined in Definition 2.*

*Proof.*   (1) Existence follows directly from Lemma 2, which shows that any global minimizer $\omega^*$ of $\mathcal{L}$ enforces

$$I(Z^*; S) > 0, \quad I(Z^*; U) = 0, \quad I(Z^*; Y \mid S) = 0. \tag{39}$$

(2) Uniqueness follows from Proposition 1: Since the objective function contains $-I(Z;S)$, the global minimizer simultaneously maximizes $I(Z;S)$, making $Z^*$ a sufficient statistic for $S$. Therefore, any other representation satisfying the three conditions must be related to $Z^*$ by an invertible map.

Together, these two facts establish full learnability of the visual IV under Definition 2. $\square$

**Remark 3.** *In practice, finite positive weights $(\alpha_1, \alpha_2, \alpha_3)$ induce a trade-off among the three mutual-information terms, so a global minimizer may allow nonzero penalties if the relevance gain outweighs them.*

## F   Pseudocode of the VIV-DG

---

**Algorithm 1** Visual Instrumental Variables for Domain Generalization (VIV-DG)

---

**Input:** Data loaders (train / val / test), device, args
**Output:** Accuracy on the test set
 1: **Initialize:** Main encoder $\mathcal{G}_\theta$, Classifier $h$, IV-learner $\mathcal{Z}_\psi$, Regressor $\mathcal{R}_\phi$
 2: Data loaders: $\mathcal{D}_{\text{train}}, \mathcal{D}_{\text{val}}, \mathcal{D}_{\text{test}}$
 3: Stage epochs: $(E_1, E_2, E_3, E_4)$, Total epochs: $E = \sum_{i=1}^{4} E_i$
 4: **for** epoch $e = 1$ to $E$ **do**
 5:     **if** $e \leq E_1$ **then**
 6:         $s \leftarrow 1$
 7:     **else if** $e \leq E_1 + E_2$ **then**
 8:         $s \leftarrow 2$
 9:     **else if** $e \leq E_1 + E_2 + E_3$ **then**
10:         $s \leftarrow 3$
11:     **else**
12:         $s \leftarrow 4$
13:     **end if**
14:     **if** $s = 1$ **or** $s = 4$ **then**                                        ▷ Stage I: Main Training
15:         Unfreeze $\mathcal{G}_\theta$, $h$; freeze $\mathcal{Z}_\psi$, $\mathcal{R}_\phi$
16:         **for** each batch in $\mathcal{D}_{\text{train}}$ **do**
17:             Compute $\mathcal{L}_{\text{causal}} = \beta_1 \mathcal{L}_{\text{cls}} + \tau \mathcal{L}_{\text{MMD}}$
18:             **if** $s = 4$ **then**
19:                 Compute $\mathcal{L}_{\text{causal}}^{+} = \beta_1 \mathcal{L}_{\text{cls}} + \beta_2 \mathcal{L}_{\text{cls}}^{\hat{s}} + \tau \mathcal{L}_{\text{MMD}}$   ▷ Stage I+: Causal Correction
20:             **end if**
21:             Backpropagate and update $\mathcal{G}_\theta$, $h$
22:         **end for**
23:     **else if** $s = 2$ **then**                                        ▷ Stage II: IV Encoder Training
24:         Freeze $\mathcal{G}_\theta$, $h$; unfreeze $\mathcal{Z}_\psi$
25:         **for** each batch in $\mathcal{D}_{\text{train}}$ **do**
26:             Compute $\mathcal{L}_{\text{total\_IV}} = \alpha_1 \mathcal{L}_{\text{MI\_ZS}} + \alpha_2 \mathcal{L}_{\text{MI\_ZU}} + \alpha_3 \mathcal{L}_{\text{MI\_ZY}}$
27:             Backpropagate and update $\mathcal{Z}_\psi$
28:         **end for**
29:     **else if** $s = 3$ **then**                                        ▷ Stage III: Regressor Training
30:         Freeze $\mathcal{G}_\theta$, $\mathcal{Z}_\psi$; unfreeze $\mathcal{R}_\phi$
31:         **for** each batch in $\mathcal{D}_{\text{train}}$ **do**
32:             Compute regression loss: $\mathcal{L}_{\text{reg}} = \mathbb{E}_{(s,z) \sim p(s,z)} \left[ \|\mathcal{R}_\phi(\text{LN}(z)) - s\|_2^2 \right]$
33:             Backpropagate and update $\mathcal{R}_\phi$
34:         **end for**
35:         **if** $e = E_1 + E_2 + E_3$ **then**
36:             Freeze $\mathcal{R}_\phi$
37:         **end if**
38:     **end if**
39: **end for**
40: Evaluate on val / test set
41: **Return:** Accuracy on the test set

---

## G   Computational cost

The computational cost of our VIV-DG is justified by the improved scalability and automation. The main computational cost of VIV-DG comes from the automatic visual IV learning module. Although this introduces additional training cost, it substantially reduces the manual effort required to design instrumental variables, which is often necessary in predefined IV-based methods. It also avoids

reliance on domain knowledge or expert heuristics. Despite achieving only moderate gains over some baselines, it consistently outperforms predefined IV-based methods like IV-DG in both performance and practicality. Therefore, we consider this a reasonable trade-off that improves both flexibility and reduces human effort. Using ResNet-50 as the backbone, training on a single A800 GPU requires approximately 40–50 GB of memory and runs stably.

The overall time complexity of our training framework is $T = \mathcal{O}(N \cdot E \cdot P_{\max})$, where $N$ denotes the number of training samples, $E$ is the total number of training epochs, and $P_{\max}$ represents the maximum computational cost among the modules trained in each stage.

## H Experimental setup

### H.1 Dataset details

We evaluate our method on four widely used domain generalization benchmarks: Digits-DG [26], PACS [27], Office-Home [28], and VLCS [29]. Detailed descriptions of these datasets are provided below:

**Digits-DG** [26] is a benchmark dataset for digit recognition, consisting of four domains: *MNIST* [48], *MNIST-M* [49], *SVHN* [50], and *SYN* [49]. These domains differ significantly in font style, background, and stroke color. Following [26], we randomly select 600 images per class in each domain, using 80% for training and 20% for validation.

**PACS** [26] is a domain generalization benchmark that includes four domains with different visual styles: *Art-Painting*, *Cartoon*, *Photo*, and *Sketch*. It contains a total of 9,991 images across 7 categories: *dog*, *elephant*, *giraffe*, *guitar*, *house*, *horse*, and *person*. For fair comparison, we use the original training-validation split provided by [27].

**Office-Home** [28] consists of four domains: *Art*, *Clipart*, *Product*, and *Real-World*. Each domain includes 65 object categories related to office and home environments, with a total of 15,588 images. Following [31], we use 90% of the data for training and 10% for validation.

**VLCS** [29] contains four domains: *Caltech*, *LabelMe*, *Sun*, and *Pascal*. Each domain shares the same 5 categories, with a total of 10,729 images.

### H.2 Baseline methods

We categorize the compared baselines into two groups: non-causal methods and causality-inspired methods. The detailed descriptions are as follows:

(i) Non-causal methods primarily rely on statistical correlations to learn domain-invariant features. FACT [7] constructs a consistency loss through data augmentation methods based on Fourier transform and co-teaching regularization. RSC [32] iteratively discards dominant features activated on training data and forces the network to activate remaining features that are correlated with labels. DeepAll [26] maps source domain training data to unseen domains via a label classifier, a domain classifier, and a domain transformation network, thereby enhancing the robustness of the label classifier to unknown domain variations. L2A-OT [35] employs a data generator to synthesize pseudo-novel domain data for augmenting the source domain, directly increasing the diversity of training domains and improving model generalization. LRDG [36] learns a domain-invariant model by tactically removing domain-specific features from the input images.

(ii) Causality-inspired approaches explicitly model causal relationships to address distribution shifts. MatchDG [34], a matching-based method, aligns inputs from the same latent object across domains to enhance out-of-domain performance. CIRL [13] utilizes data augmentation techniques and causal intervention methods to learn causal representations. IV-DG [17] leverages instrumental variables to eliminate confounding effects and considers that data from one domain can serve as instrumental variables for another domain. FAGT [16] eliminates confounding effects through style transfer and the front-door adjustment method. iDAG [41] extracts invariant graph structures as proxies for causal structures to enhance representation generalization. GMDG [43] jointly learns domain-invariant conditional features and maximizes the posterior, providing a flexible framework that generalizes and theoretically explains existing multi-domain generalization methods.

### H.3 Implementation details

#### H.3.1 Basic details

In our experiments, we used an NVIDIA A800 80GB PCIe GPU. We adopt the same backbone for both the primary encoder and the instrumental variable encoder across all benchmarks. For Digits-DG, we follow the architecture used in [35], training from scratch with a batch size of 8 and an initial learning rate of 0.001, which decays by a factor of 0.1 after 80% of the total epochs. For PACS and Office-Home, we use ImageNet-pretrained ResNet-18 and ResNet-50 [51] backbones with a batch size of 5. VLCS follows the same configuration using ResNet-50. All models are trained using SGD optimizer with a momentum of 0.9. All reported results are averaged over 3 runs. Following the DomainBed protocol, we adopt the official data splits, model selection strategy, and evaluation settings to ensure the fairest possible comparisons. Except for the batch size constrained by GPU memory and the model-specific training epochs, other configurations, including dataset splits, evaluation protocols, backbone architectures, and the leave-one-domain-out strategy, remain identical to the DomainBed defaults. For fair comparison, our main extraction network (initial training and bias correction fine-tuning) is trained for 50 epochs, consistent with the compared methods.

#### H.3.2 Method-specific details

We design a stage-wise training scheme for both VIV-DG and VIV-DG-Lite, with detailed configurations summarized in Table 4 and Table 5. Notably, the main encoder $\mathcal{G}_\theta$ is trained for a total of 50 epochs, consisting of the initial training in Phase I and the fine-tuning stage in Phase I+.

Table 4: Epoch configuration for stage-wise training of VIV-DG

| Stage | Training Objective | Digits-DG | PACS (ResNet-18) | Office-Home (ResNet-18) |
|---|---|---|---|---|
| Phase I | Train main encoder $\mathcal{G}_\theta$ | 30 | 30 | 30 |
| Phase II | Freeze $\mathcal{G}_\theta$; train IV encoder $\mathcal{Z}_\psi$ | 25 | 25 | 25 |
| Phase III | Freeze $\mathcal{G}_\theta$, $\mathcal{Z}_\psi$; train regressor $\mathcal{R}_\phi$ | 25 | 25 | 25 |
| Phase I+ | Freeze $\mathcal{Z}_\psi$, $\mathcal{R}_\phi$; fine-tune $\mathcal{G}_\theta$ | 20 | 20 | 20 |

† Phase I+ refers to a fine-tuning stage built upon Phase I.

Table 5: Epoch configuration for stage-wise training of VIV-DG-Lite

| Stage | Training Objective | VLCS (ResNet-50) | PACS (ResNet-50) | Office-Home (ResNet-50) |
|---|---|---|---|---|
| Phase I | Train main encoder $\mathcal{G}_\theta$ | 30 | 30 | 30 |
| Phase II | Freeze $\mathcal{G}_\theta$; train IV encoder $\mathcal{Z}_\psi$ | 20 | 20 | 20 |
| Phase I+ | Freeze $\mathcal{Z}_\psi$; fine-tune $\mathcal{G}_\theta$ | 20 | 20 | 20 |

† Phase I+ refers to a fine-tuning stage built upon Phase I.

#### H.3.3 Hyperparameter settings

The hyperparameter settings are summarized in Table 6, encompassing the various datasets and training stages employed in our experiments.

**Phase I (Initial causal factor extraction)**: The loss function $\mathcal{L}_{\mathrm{causal}}$ incorporates two hyperparameters: $\beta_1$ and $\tau$. $\beta_1$ is set per dataset (Digits-DG = 1, PACS = 0.5, Office-Home = 0.1, VLCS = 0.1), while $\tau$ adapts dynamically according to the number of training epochs. This scheme ensures rapid convergence across datasets and stabilizes causal factor extraction in early training.

**Phase II (Visual IV learning) & Phase III (Regressor training)**: To reduce tuning complexity, Phases II and III adopt identical hyperparameter settings across all datasets. Specifically, in Phase II's total loss $\mathcal{L}_{\mathrm{total\_IV}}$, we set $\alpha_1 = 1$, $\alpha_2 = 0.5$, and $\alpha_3 = 0.5$. In Phase III, the regressor loss $\mathcal{L}_{\mathrm{reg}}$ carries a default weight of 1 to maintain stability.

**Phase I+ (Causal factor refinement and debiasing)**: The loss $\mathcal{L}_{\mathrm{causal}}^+$ reuses $\beta_1$ and $\tau$ from Phase I while introducing a dynamically adjusted $\beta_2$. This adaptive strategy enables the model to flexibly respond to distributional shifts and further refine debiasing.

Table 6: Hyperparameter settings

| Stage | Loss Function | Hyperparameters | Settings |
|---|---|---|---|
| Phase I | $\mathcal{L}_{\text{causal}}$ | $\beta_1, \tau$ | $\beta_1$: Digits-DG=1; PACS=0.5; Office-Home=0.1; VLCS=0.1; $\tau$: adaptively adjusted |
| Phase II | $\mathcal{L}_{\text{total\_IV}}$ | $\alpha_1, \alpha_2, \alpha_3$ | $\alpha_1 = 1; \alpha_2 = 0.5; \alpha_3 = 0.5$ |
| Phase III | $\mathcal{L}_{\text{reg}}$ | — | Regression weight set to 1 by default |
| Phase I+ | $\mathcal{L}_{\text{causal}}^+$ | $\beta_1, \beta_2, \tau$ | $\beta_1, \tau$: same as Phase I; $\beta_2$: adaptively adjusted |

[†] Phase I+ refers to a fine-tuning stage built upon Phase I.

### H.3.4 Stage-wise training configuration of VIV-DG and VIV-DG-Lite

Table 7 illustrates the stage-wise training procedures for both VIV-DG and its simplified variant, VIV-DG-Lite. Notably, Phase III is omitted in VIV-DG-Lite, as the regressor training is replaced by a feature fusion strategy, as described in Remark 2.

Table 7: Stage-wise training configuration of VIV-DG and VIV-DG-Lite

| Stage | Training Objective | VIV-DG | VIV-DG-Lite |
|---|---|---|---|
| Phase I | Train main encoder $\mathcal{G}_\theta$ | ✓ | ✓ |
| Phase II | Freeze $\mathcal{G}_\theta$; train IV learner $\mathcal{Z}_\psi$ | ✓ | ✓ |
| Phase III | Freeze $\mathcal{G}_\theta, \mathcal{Z}_\psi$; train regressor $\mathcal{R}_\phi$ | ✓ | – |
| Phase I+ | Freeze $\mathcal{Z}_\psi$ and optionally $\mathcal{R}_\phi$; fine-tune $\mathcal{G}_\theta$ | ✓ | ✓ |

[†] ✓ indicates that the phase is included in the method. Phase I+ refers to a fine-tuning stage based on Phase I.

### H.4 Hyperparameter sensitivity analysis design

Table 8 summarizes the value ranges and step sizes of the hyperparameters involved in each training stage for sensitivity analysis.

Table 8: Hyperparameter sensitivity analysis design

| Parameter(s) | Value Range (Step Size) | Default Value |
|---|---|---|
| **Single-variable Sensitivity** | | |
| $\beta_1$ | {0.1, 0.3, 0.5, 0.7, 0.9} (linear) | 0.5 |
| $\alpha_1$ | {0.4, 0.7, 1.0, 1.3, 1.6} (linear) | 1.0 |
| $\alpha_2$ | {0.1, 0.3, 0.5, 0.7, 0.9} (linear) | 0.5 |
| $\alpha_3$ | {0.1, 0.3, 0.5, 0.7, 0.9} (linear) | 0.5 |
| **Multi-variable Combinations of $\alpha_1, \alpha_2,$ and $\alpha_3$** | | |
| | C1: (0.5, 1.0, 0.5) | 1.0 / 0.5 / 0.5 |
| | C2: (1.0, 0.2, 0.2) | 1.0 / 0.5 / 0.5 |
| $\alpha_1 + \alpha_2 + \alpha_3$ | C3: (1.0, 0.5, 0.5) | 1.0 / 0.5 / 0.5 |
| | C4: (0.5, 0.5, 1.0) | 1.0 / 0.5 / 0.5 |
| | C5: (1.0, 0.3, 0.7) | 1.0 / 0.5 / 0.5 |

## I Additional results

### I.1 Evaluation on PACS and Office-Home using ResNet-50

Table 9 and Table 10 present the experimental results on PACS and Office-Home using ResNet-50, respectively. These results further demonstrate the effectiveness of our method, and in particular, the results in Table 10 show that VIV-DG exhibits significant superiority on Office-Home.

Table 9: The results on PACS with ResNet-50

| Methods | A | C | P | S | Avg. |
|---|---|---|---|---|---|
| SagNet [39] | 87.4 | 80.7 | 97.1 | 80.0 | 86.3 |
| MatchDG [34] | 85.6 | 82.1 | 97.9 | 78.8 | 86.1 |
| SAGM [52] | 87.4 | 80.2 | **98.0** | 80.8 | 86.6 |
| CIRL [13] | 90.7 | **84.3** | 97.8 | **87.7** | **90.1** |
| FACT [7] | **90.9** | 83.7 | 97.8 | 86.2 | 89.7 |
| RICE [42] | 87.8 | **84.3** | 96.8 | 84.7 | 88.4 |
| iDAG [41] | 90.8 | 83.7 | **98.0** | 82.7 | 88.8 |
| GMDG [43] | 84.7 | 81.7 | 97.5 | 80.5 | 85.6 |
| VIV-DG-Lite | 90.3 | 83.8 | 97.6 | 86.7 | 89.6 |
| VIV-DG | 90.5 | 83.6 | 97.8 | 86.7 | 89.7 |

† The best and second best results are marked in bold and underlined, respectively. Avg. = Average accuracy (%).

Table 10: The results on Office-Home with ResNet-50

| Methods | A | C | P | R | Avg. |
|---|---|---|---|---|---|
| RSC [32] | 60.7 | 51.4 | 74.8 | 75.1 | 65.5 |
| MixStyle [38] | 51.1 | 53.2 | 68.2 | 69.2 | 60.4 |
| SagNet [39] | 63.4 | 54.8 | 75.8 | 78.3 | 68.1 |
| PCL [40] | 67.3 | 59.9 | 78.7 | 80.7 | 71.6 |
| SAGM [52] | 65.4 | 57.0 | 78.0 | 80.0 | 70.1 |
| FAGT [16] | 66.3 | 59.5 | 77.6 | 79.0 | 70.6 |
| iDAG [41] | 68.2 | 57.9 | 79.7 | 81.4 | 71.8 |
| GMDG [43] | **68.9** | 56.2 | **79.9** | **82.0** | 70.7 |
| VIV-DG-Lite | 67.8 | 60.7 | 79.7 | 81.4 | 72.4 |
| VIV-DG | 68.4 | **62.3** | 79.4 | 81.7 | **73.0** |

† The best and second best results are marked in bold and underlined, respectively. Avg. = Average accuracy (%).

## I.2 Sensitivity to mask threshold $\delta$

To verify the robustness of the parameter threshold $\delta$ used in the mask generation process (Sec. D), we conduct sensitivity experiments on $\delta$ for VIV-DG using the Digits-DG and PACS datasets, with PACS experiments conducted using the ResNet-18 backbone. The results are shown in Figure 9. We vary $\delta$ from 20 to 40 in steps of 5 (20, 25, 30, 35, and 40) and observe minimal performance variation, indicating that VIV-DG is robust with respect to the parameter $\delta$.

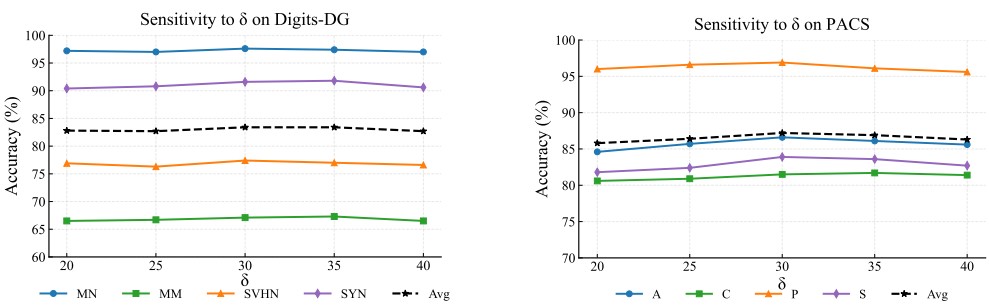

Figure 9: Sensitivity analysis of mask threshold $\delta$: (Left) Digits-DG; (Right) PACS.

## I.3 Visual explanation

The visualization results on the PACS dataset, with *Cartoon*, *Photo*, and *Sketch* as the target domains, are shown in Figure 10, Figure 11, and Figure 12, respectively. It can be observed that VIV-DG attends to broader and more holistic discriminative regions, whereas CIRL [13] tends to rely on more localized cues. This demonstrates the effectiveness of our method in learning causal representations.

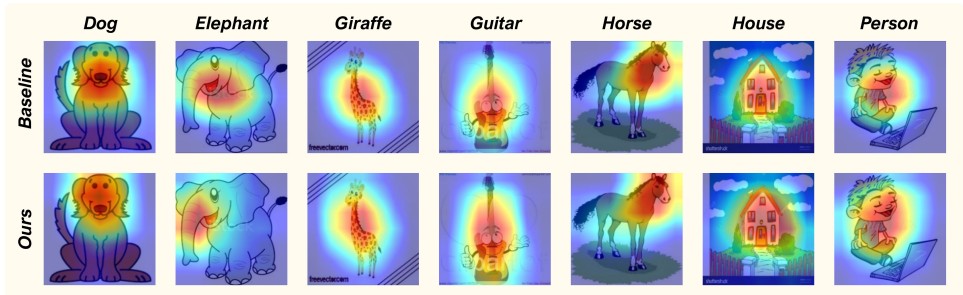

Figure 10: Grad-CAM visualization on the PACS dataset with *Cartoon* as the target domain.

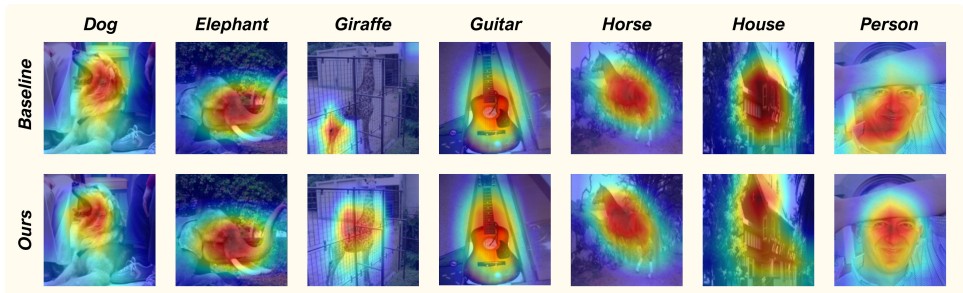

Figure 11: Grad-CAM visualization on the PACS dataset with *Photo* as the target domain.

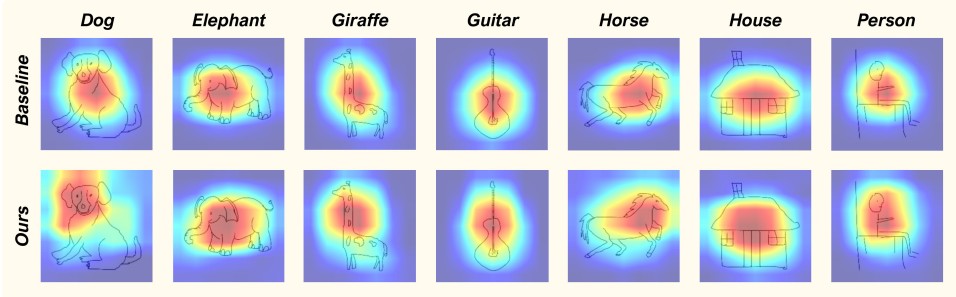

Figure 12: Grad-CAM visualization on the PACS dataset with *Sketch* as the target domain.

## I.4 Mutual information analysis

To verify whether the representations learned by the visual instrumental variable learner satisfy the instrumental variable conditions, we estimate three key mutual information terms $I(Z;S)$, $I(Z;U)$, and $I(Z;Y|S)$ on the PACS dataset. For better interpretability and consistent scaling, all mutual information values are normalized to the range [0, 1]. As shown in Figure 13, the results indicate that the learned representation $Z$ exhibits high mutual information with the causal factor $S$, low mutual information with the confounder $U$, and low conditional mutual information

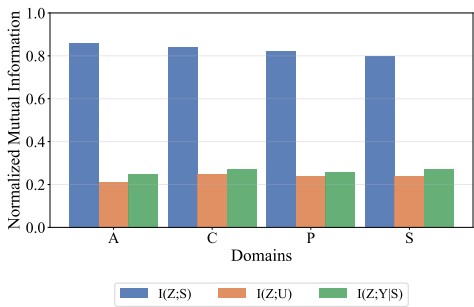

Figure 13: Mutual information analysis.

with the potential outcome $Y$ given $S$. These findings demonstrate that the learned Visual IVs satisfy the relevance, exclusion, and independence constraints, thereby validating their effectiveness.

## I.5 Convergence analysis

We analyze the convergence behavior on the *Art-Painting* domain, with the corresponding curves presented in Figure 14 and Figure 15. Specifically, Figure 14 demonstrates that the loss decreases steadily across all stages. Figure 15 reveals that VIV-DG converges rapidly during the early phase of causal representation learning, with accuracy further improving in Phase I+ (causal correction and debiasing) compared to Phase I. Notably, during Phase II and Phase III, while training the instrumental variable learner and the regressor, the main network parameters remain unchanged, so the accuracy shown in Figure 15 for these two phases stays the same as in the final epoch of Phase I.

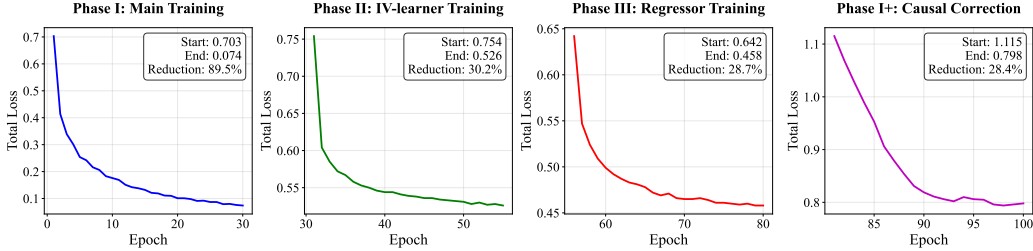

Figure 14: Total loss convergence by stage.

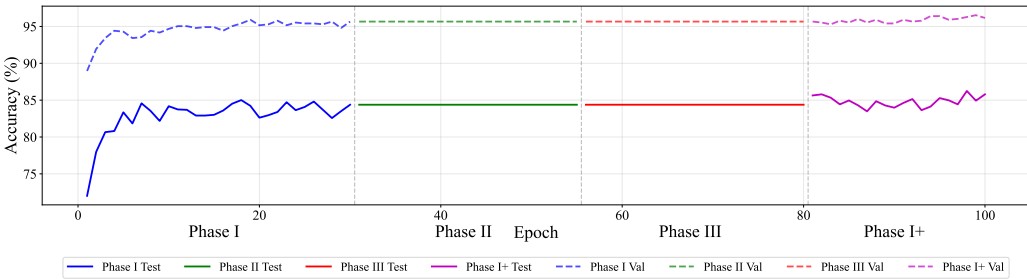

Figure 15: Overall test/validation accuracy convergence.

## J  Broader impact

Our work leverages the exogeneity of visual instrumental variables to mitigate confounding effects, achieving robust domain generalization even in the presence of unobserved confounders, which demonstrates significant societal value. By mitigating confounding effects, it enhances model robustness in open environments, reducing the influence of environmental changes on classification results and ensuring more reliable performance in real-world image classification tasks. Furthermore, the improved cross-domain generalization capability facilitates broader applications in medical imaging, autonomous driving, and security surveillance, thereby amplifying the technology's societal value. However, this research may also involve certain potential risks that warrant further academic investigation. Specifically, the reliability of the system's output could be undermined when deployed in real-world scenarios that diverge significantly from the training distribution.

