# OpenReview forum: "Automatic Visual Instrumental Variable Learning for Confounding-Resistant Domain Generalization"
_NeurIPS.cc/2025/Conference — NeurIPS 2025 poster_

### Official Review · Reviewer_z3cX · 2025-07-02

**Clarity:** 3
**Significance:** 2
**Originality:** 2
**Rating:** 2
**Confidence:** 5

**Summary:**

This work proposes Visual Instrumental Variable Learning for confounding-resistant domain generalization, which introduces instrumental variables to address the challenge of identifying confounders. The optimization process is decoupled into three stages:
(i) The first stage utilizes frequency-based augmentation to learn causal factors under the constraints of MMD and classification loss;
(ii) The second stage aims to learn implicit visual instrumental variables using three designed loss functions;
(iii) The third stage trains a regressor to learn causal factors responsible for classification.
Experiments are conducted on PACS, VLCS, Office-Home, and Digits-DG.

**Questions:**

1. Why is VIV-DG not included in Table 2?
2. How are the positive and negative samples generated? How are the confounder indices identified in Sections 3.4.2 and 3.4.3?

**Ethical Concerns:**

["NO or VERY MINOR ethics concerns only"]

**Final Justification:**

During the rebuttal phase, the authors still have not provided experiments on additional benchmarks from DomainBed, which makes it difficult to fully demonstrate the effectiveness of their approach. DomainBed includes more than the three datasets used in the paper.
In W5, the visualization cannot demonstrate the IV and causal features effectively. Furthermore, the response to the point, "However, these factors do not appear in the current observed input data or known source domains, and therefore do not exert a direct causal effect on the input data," is not accurate. In the causal graph, there must be a connection between C and the input. The current explanation does not address this issue appropriately. Therefore, I maintain my original score

**Limitations:**

yes

**Quality:**

2

**Strengths And Weaknesses:**

Strengths:
(i) The proposed visual instrumental variable learning is theoretically effective and interesting for causal representation learning.
(ii) The design of the three loss functions adheres to the principles of instrumental variable learning.
(iii) The experiments outperform some existing methods.
Weaknesses: There are several issues that need to be addressed:
(i) Although the theory is interesting, the modeling may not be appropriate. The causal graph (i.e., Figure 4) appears incorrect—why do unobserved confounders not affect the input data? Although the appendix provides an explanation for the partial arrows in the figure, further clarification and examples are needed, as causal theory is inherently abstract.
(ii) The experimental results are not convincing. The method is not evaluated on the standard DomainBed benchmark, and comparisons with more other causal learning-based methods are required.
(iii) The optimization process is complex. How does its time complexity compare with other methods?
(iv) The example involving only color and texture may not be sufficient to identify confounders. More comprehensive examples are needed.
(v) The paper should clarify what can be visualized in the original figure or visualize the features learned for the instrumental variable and causal features SSS.
(vi) The ablation studies are simple and lack sufficient investigation into the three designed loss functions.

---

> ### Author Rebuttal · Authors · 2025-07-31
>
> We are grateful for your recognition of the theoretical effectiveness and conceptual interest of our proposed visual IV learning framework. Here, we clarify and address these issues accordingly.
>
> **Questions:**
>
> **Q1: Why is VIV-DG not included in Table 2?**
>
> **R1:** We thank the reviewer for pointing this out. This was our oversight. We now supplement the results of VIV-DG on the VLCS dataset, further demonstrating the effectiveness of VIV-DG.
> #### **Tab. 1 Results of VIV-DG on VLCS**
> |Methods|C|L|S|P|Avg.|
> |-|-|-|-|-|-|
> |VIV-DG-Lite | **99.2** | 67.4 | 74.4 | **77.6**|**79.7**|
> |VIV-DG|99.0 |**67.7** |**74.6** |77.5|**79.7**|
>
> **Q2: Positive and negative samples. Confounder indices.**
>
> **R1: (1) Explanation of positive and negative samples generated.** In Section 3.4.2, positive sample pairs consist of visual instrumental variables $Z$ and causal representations $S$ from the same original sample, denoted as ${(z_i, s_i)}_{i=1}^N$, where both $z_i$ and $s_i$ originate from the image $x_i$. Negative sample pairs are formed by combining $Z$ and $S$ from different samples, $(z_i, s_j)$, where $i \neq j$, created by cross-sample indexing. Similarly, in Section 3.4.3, positive sample pairs consist of $Z$ and confounders $U$ from the same image, while negative sample pairs combine $Z$ with shuffled confounders $U$ from different samples. This construction of positive and negative samples effectively approximates mutual information estimation during mini-batch training.
>
> **(2) Explanation of confounder indices:** The indices of confounders are automatically identified and separated by the model through the positive-negative sample contrastive mechanism during training, rather than being explicitly annotated or manually defined.
>
> **Weaknesses:**
>
> **W1: Explanation of the rationality of the causal graph.**
>
> **R1:** We apologize for the misunderstanding caused by our unclear explanation and appreciate the opportunity to clarify. In Fig. 4, the unobserved confounder $C$ does not have a direct causal link to the input image $X$, which is an intentional modeling choice. Without loss of generality, we assume that $C$ represents confounders related to **unknown domain** styles or backgrounds. Since domains cannot be exhaustively enumerated, such unobserved confounders inevitably exist. However, these factors do not appear in the current observed input data or known source domains, and therefore do not exert a direct causal effect on the input data $X$. Conversely, adding a direct edge from $C$ to $X$ would introduce ambiguity by implying that $C$ is the observed confounder.
>
> It is worth noting that both known and unknown domains share the same causal features $S$ and potential outcome $Y$. In the unseen domain where the confounder $C$ exists, $C$ also affects $S$ and $Y$. Therefore, we model this by adding causal edges from $C$ to both $S$ and $Y$ in Fig. 4.
>
>
> **W2: Not evaluated on DomainBed and comparisons with other causal methods.**
>
> **R1: (1) Standard DomainBed benchmark:** Although we do not run our experiments directly on the DomainBed codebase, our experimental setup strictly follows the standard DomainBed protocol, including dataset splits, backbones, and the leave-one-domain-out strategy. For fair comparison，our main extraction network (initial training + bias correction fine-tuning) is trained for 50 epochs, consistent with the compared methods.
>
> **(2) More causal methods:** We further investigated and included several causal methods. The results are as follows. Our method consistently outperforms these approaches.
>
> #### **Tab. 2 Results on Digits-DG**
> |Methods|MN|MM|SVHN|SYN|Avg.|
> |-|-|-|-|-|-|
> |CDIM[1]|98.7|64.0|74.1|92.9|82.4|
> |VIV-DG|97.6|67.1|77.4|91.6|83.4|
>
> #### **Tab. 3 Results on PACS (ResNet18)**
> |Methods|A|C|P|S|Avg.|
> |-|-|-|-|-|-|
> |CDIM[1]|83.6|77.6|95.5|78.2|83.7|
> |VIV-DG|86.6|81.5|96.9|83.9|**87.2**|
>
> #### **Tab. 4 Results on PACS (ResNet50)**
> |Methods|A|C|P|S|Avg.|
> |-|-|-|-|-|-|
> |CSRDN[2]|88.3|84.5|97.5|84.9|88.8|
> |RICE[3]|87.8|84.3|96.8|84.7|88.4|
> |VIV-DG-Lite|90.3|83.8|97.6|86.7|**89.6**|
>
> #### **Tab. 5 Results on VLCS (ResNet50)**
> |Methods|C|L|S|P|Avg.|
> |-|-|-|-|-|-|
> |CSRDN[2]|98.8|67.0|74.2|77.1|79.3|
> |RICE[3]|98.3|69.2|74.6|75.1|79.3|
> |VIV-DG-Lite | 99.2 | 67.4 | 74.4 | 77.6|**79.7**|
> |VIV-DG|99.0 |67.7 |74.6 |77.5|**79.7**|
>
> [1] Unbiased Semantic Representation Learning Based on Causal Disentanglement for Domain Generalization. (2024)
>
> [2] A Causality-Aware Perspective on Domain Generalization via Domain Intervention. (2024)
>
> [3] Out-of-distribution Generalization with Causal Invariant Transformations. (2022)
>
> **W3: Time complexity.**
>
> **R3: (1) Computational Complexity:** The overall time complexity of our training framework is: $T = \mathcal{O}(N \cdot E \cdot P_{\max})$ where $N$ denotes the number of training samples, $E$ is the total number of training epochs, and $P_{\max}$ represents the maximum computational cost among the modules trained in each stage. This complexity does not involve exponential or polynomial growth, making it computationally tractable.
>
> **(2) The primary computational cost arises from the automatic visual IV learning module. However, we consider this a reasonable and acceptable trade-off, given the improvements in automation and generalization capability that it brings.**  Specifically, the training process of VIV-DG is slightly more complex and time-consuming than some baseline methods, mainly due to its multi-stage Visual IV learning. However, unlike predefined IV methods, our approach does not rely on domain experts for IV design and selection, avoiding high human costs and prior bias. It also offers better adaptability and scalability.
>
> **W4: More comprehensive examples.**
>
> **R4:** We appreciate the suggestion and will include more representative examples of confounders in the revised version, such as scene background, co-occurrence bias, image style, lighting conditions, and camera angles. They are typically entangled and difficult to disentangle in the absence of labeled supervision. These confounders tend to exhibit similar patterns in the Fourier amplitude information of images, such as color, style, and brightness. Therefore, we propose a feature extraction method based on reconstructing images from amplitude information to model the confounders. Our Visual IV method mitigates the influence of both observed and unobserved confounders without explicitly modeling them.
>
> **W5: The paper should clarify what can be visualized in the original figure or visualize the features learned for the IV and causal features.**
>
> **R5:** We apologize for the unclear explanation of the visualization experiments, and we clarify the details as follows. We compare our VIV-DG with CIRL, a strong causal domain generalization baseline, in the visualization experiments. We intentionally select the *Art Painting* domain—where the classification accuracies of the two methods are similar—to demonstrate that our method better captures causal regions. Even if the improvement is not visually striking, it still shows progress in identifying causal cues.
>
> To further demonstrate the causal representation learning ability of our method, we extend the Grad-CAM visualizations to the *Cartoon* and *Sketch* domains. (We cannot display the images here.) The results further show that VIV-DG focuses more on the central structure and semantically relevant object regions, which is consistent with the superior performance of VIV-DG over CIRL in these domains.
>
> **Additional visualization evidence:** To verify that the learned instrumental variables alleviate confounding effects, we visualize the Grad-CAM maps of causal features before and after IV-based correction. The visual results show that the model captures more robust and semantically meaningful regions after using the visual IVs.
>
> **Quantitative verification through mutual information:** Since visual results cannot be displayed here, we also provide more direct and quantitative evidence of IV effectiveness. We compute the estimated mutual information values of $I(Z;S)$, $I(Z;U)$, and $I(Y;Z|S)$ to verify whether the learned representation $Z$ satisfies the relevance, independence, and exclusion conditions of an instrumental variable. The results indicate that the learned representation approximately satisfies the IV conditions and thus serves as an effective guidance for mitigating confounding bias. We will include these results in the revised version.
> #### **Tab. 6 Estimated values of mutual information on PACS**
> ||A|C|
> |-|-|-|
> |I(Z;S)|0.86|0.84|
> |I(Z;U)|0.21|0.25|
> |I(Z;Y\|S)|0.25|0.27|
>
> **W6: Ablation studies.**
>
> **R6:** We supplement ablation studies on the three mutual information loss terms (**relevance**, **independence**, and **exclusion**) for both versions of our model, VIV-DG and VIV-DG-Lite, on the PACS dataset, as shown below.
>
> #### **Tab. 7 Ablation study of VIV-DG on PACS (ResNet18)**
> | Settings| A|C|P|S| Avg.|
> |-|-|-|-|-|-|
> |w/o I(Z;S)|85.5|80.4|95.8|83.6|86.3|
> |w/o I(Z;U)|84.9|79.7|95.9|83.0|85.9|
> |w/o I(Z;Y\|S) |84.6|79.1|95.3|82.4|85.4|
> |VIV-DG |86.6|81.5|96.9|83.9|87.2|
>
> #### **Tab. 8 Ablation study of VIV-DG-Lite on PACS (ResNet18)**
> |Settings| A|C|P|S| Avg.|
> |-|-|-|-|-|-|
> |w/o I(Z;S)|85.6|79.7|95.4|82.9|85.9|
> |w/o I(Z;U)|84.7|78.4|95.5|82.4|85.3|
> |w/o I(Z;Y\|S) |84.5|78.2|95.5|81.9|85.0|
> |VIV-DG-Lite|86.1|81.8|96.6|83.3|87.0|
>
> The results show that the **exclusion** loss is the most critical, as it effectively prevents the introduction of extra confounders as instrumental variables. Meanwhile, the **relevance** and **independence** losses work together to reduce confounding effects and enhance the expressiveness of causal representations. These three loss terms collaboratively contribute to the strong performance of both the full and Lite versions. These results, together with those from other datasets, will be included in the revised version.

---

> > ### Comment · Reviewer_z3cX · 2025-08-06
> > **feedback**
> >
> > thanks for the rebuttal comments, since not all concerns are addressed, I will keep the score.

---

### Official Review · Reviewer_ubcT · 2025-07-02

**Clarity:** 3
**Significance:** 3
**Originality:** 2
**Rating:** 4
**Confidence:** 4

**Summary:**

This paper proposes a novel confounding-resistant DG method termed VIV-DG. The VIV-DG automatically learns visual instrumental variables to effectively mitigate the effects of both observed and unobserved confounders and improve domain generalization. However, obtaining valid IVs remains a challenge. The paper defines the novel concept of visual instrumental variables and develops an IVs learning framework composed of three alternately optimized subnetworks that automatically learns valid ones.  Extensive experiments verify the effectiveness of VIV-DG on generalization.

**Questions:**

See the Weaknesses.

**Ethical Concerns:**

["NO or VERY MINOR ethics concerns only"]

**Final Justification:**

I have already read all the comments and discussions about this paper and have finally decided to keep my initial score.

**Limitations:**

yes

**Quality:**

2

**Strengths And Weaknesses:**

Strengths:
1. The paper is well-structured, and its concepts are clearly defined.
2. The motivation is clear. Considering the intrinsic differences among non-casual factors and identifying the valid instrumental variables to improve generalization is reasonable and promising.
3. The visual explanation well illustrated the effectiveness of VIV-DG in capturing the core object structures.

Weaknesses:
1. The ablation study is not complete. The ablation study should provide a more in-depth analysis of each component in Visual IV learning and the regression predictor, such as relevance-constrained learning.
2. Missing experimental results on other common backbones, such as Vision Transformer.
3. Missing classification experiments on large-scale datasets (such as ImageNet), as well as experiments on other OOD generalization datasets like (ImageNet-Sketch).
4. Missing theoretical proofs of the VIV-DG, as well as analysis and proofs of the generalization's lower and upper bounds.
5. Missing experiments and analysis of the VIV-DG 's efficiency.

---

> ### Author Rebuttal · Authors · 2025-07-31
>
> We sincerely appreciate your support of the novelty of our work and your valuable comments. Here, we clarify and address these issues accordingly.
>
> **Q1: Ablation study:**
>
> **R1: (1) Ablation studies on the three mutual information loss:** We supplement ablation studies on the three mutual information loss terms (**relevance**, **independence**, and **exclusion**) for both versions of our model, VIV-DG and VIV-DG-Lite, on the PACS dataset, as shown below.
>
> The results show that the exclusion loss is the most critical, as it effectively prevents the introduction of extra confounders as IVs. Meanwhile, the relevance and independence losses help reduce confounding effects and enhance the expressiveness of causal representations. These results, together with those from other datasets, will be included in the revised version.
> #### **Tab. 1 Ablation study of VIV-DG on PACS (ResNet18)**
> | Settings| A|C|P|S| Avg.|
> |-----------------|-------|-------|-------|---------|--------|
> |w/o I(Z;S)|85.5|80.4|95.8|83.6|86.3|
> |w/o I(Z;U)|84.9|79.7|95.9|83.0|85.9|
> |w/o I(Z;Y\|S) |84.6|79.1|95.3|82.4|85.4|
> |VIV-DG |86.6|81.5|96.9|83.9|87.2|
>
> #### **Tab. 2 Ablation study of VIV-DG-Lite on PACS (ResNet18)**
> | Settings| A|C|P|S| Avg.|
> |-----------------|-------|-------|-------|---------|--------|
> |w/o I(Z;S)|85.6|79.7|95.4|82.9|85.9|
> |w/o I(Z;U)|84.7|78.4|95.5|82.4|85.3|
> |w/o I(Z;Y\|S) |84.5|78.2|95.5|81.9|85.0|
> |VIV-DG-Lite|86.1|81.8|96.6|83.3|87.0|
>
> **(2) Ablation analysis of the regression predictor:**
> The ablation analysis of the regression module is essentially reflected in the comparison between VIV-DG and VIV-DG-Lite: VIV-DG incorporates a regressor to predict causal representations, while VIV-DG-Lite does not. As shown in Table 1 of the paper, the performance of VIV-DG and VIV-DG-Lite varies across target domains. We provide further analysis to explain this: **Overall, the regression predictor effectively prevents the learned causal representations from being affected by confounders.**
>
> Specifically, VIV-DG-Lite approximates the causal representations by combining the instrumental variables with the initial causal representations, which retains more original discriminative information but is more susceptible to confounding. Therefore, it performs better in domains with smaller distribution shifts, such as *Product* in Office-Home and *Cartoon* in PACS dataset.
>
> In contrast, VIV-DG predicts causal representations through a trained regressor for bias correction, which helps reduce confounding effects and thus performs better in domains with larger distribution shifts, such as *Sketch* in PACS dataset. Overall, VIV-DG demonstrates greater robustness compared to VIV-DG-Lite.
>
> **Q2: Missing experimental results on other common backbones.**
>
> **R2:** The baseline methods we compare with mainly use ResNet18 and ResNet50. For fair comparison, we adopt the same backbones and do not conduct experiments on Vision Transformer. In fact, we use three different backbones in total: ResNet18, ResNet50, and a lightweight backbone with the same architecture as existing works on the Digits-DG dataset.
>
> In addition, *Lemma 2* and the *Theorem* provided in our response to Question 4 support that our method can automatically learn instrumental variables across different backbone architectures.
>
> **Q3: Experiments on large-scale datasets.**
>
> **R3:** We supplement experiments on the very large-scale domain generalization dataset DomainNet (586,575 images, 345 classes, and 6 domains). Due to the dataset’s size and long training time, we currently report results with *Sketch* as the target domain where distribution shift is large, and the other five domains as source domains. The results are as follows:
>
> #### **Tab. 3 Results on DomainNet (ResNet50)**
> ||SagNet |RSC |iDAG|VIV-DG|
> |-----------------|-------|-------|------|------|
> |Sketch |48.8|47.8|56.9|56.9|
>
> **Q4: Theoretical proofs of the VIV-DG.**
>
> **R4:** We supplement a theoretical analysis to demonstrate both the learnability and identifiability of Visual IVs, thereby establishing the effectiveness and theoretical soundness of our VIV-DG approach. Below we summarize the main theoretical results and proof ideas; full details will be included in the revised version:
>
> - **Lemma 1 (Identifiability Conditions for Visual IVs via (Conditional) Mutual Information):** Let $(S, U, Y)$ be three random variables, where
> $S$ denotes the causal factor affecting $Y$,
> $U$ is the confounder,
> and $Y$ is the downstream label.
> A candidate variable $Z$ is called a visual instrument variable if it simultaneously satisfies: (i) Relevance: $I(Z;S)>0$; (ii) Independence: $I(Z;U)=0$; (iii) Exclusion: $I(Z;Y\mid S)=0$. Here $I(\cdot;\cdot)$ and $I(\cdot;\cdot\mid\cdot)$ denote (conditional) mutual information.
>
> **Proof idea:** Based on the integral definition of mutual information, we establish the connection between IV conditions and mutual information. Each condition is verified individually to show that the three requirements for IVs can be rigorously expressed in terms of mutual information. We demonstrate that the three classical IV conditions—relevance, independence, and exclusion—can be rigorously characterized using (conditional) mutual information.
>
> - **Lemma 2 (Learnability of Visual IVs):** Let $h_\psi:\mathrm{Image}\to Z$ be a parameterized mapping (e.g., via a deep network), with the following training objective
> \begin{equation}
> \mathcal{L}(\psi)
> \=\
> -\alpha_1 I(Z;S\bigr)
> \+\\alpha_2 I(Z;U\bigr)
> \+\\alpha_3 I(Z;Y\mid S\bigr),
> \end{equation}
> where $\alpha_1,\alpha_2, \alpha_3 >0$.  Then any global minimizer $\psi^*$ satisfies
>
> $I(h_{\psi^*}(\mathrm{Image});S)>0$,
>
> $I(h_{\psi^*}(\mathrm{Image});U)>0$,
>
> $I(h_{\psi^*}(\mathrm{Image});\,Y\mid S)>0$,
> i.e. it exactly recovers a Visual IVs.
>
> **Proof idea:** We define an objective function that combines the three mutual information terms, and prove that any global minimizer must satisfy all three conditions, thereby yielding a valid Visual IV. Since $-I(Z;S)\ge -\sup_\psi I(Z;S)$,  $I(Z;U)\ge0$,  $I(Z;Y\mid S)\ge0$, any global minimizer $\psi^*$ must simultaneously (i) maximize $I(Z;S)$, (ii) drive $I(Z;U)$ to zero, and (iii) drive $I(Z;Y\mid S)$ to zero; otherwise one could reduce $\mathcal{L}$ further. By the zero‐mutual‐information conditions we get $Z\perp\ U$ and $Z\perp\ Y\mid S$, and by the maximization $I(Z;S)>0$. These three facts are exactly the three requirements of the definition of Visual IVs.
>
> - **Theorem (Identifiability for Visual IVs):** Let $(S, U, Y)$ be three random variables, where
> $S$ denotes the causal factor affecting $Y$,
> $U$ is the confounder,
> and $Y$ is the downstream label. Assume that $\mathcal H_{Z}=\{h_\psi:\mathrm{Image}\to Z\}$  is a sufficiently expressive family of mappings (e.g., one that contains all smooth bijections on a latent subspace).
> The optimize the objective
> \begin{equation}
> \mathcal{L}(\psi)
> \=\
> -\alpha_1 I(h_\psi(\mathrm{Image});S\bigr)
> \+\\alpha_2 I(h_\psi(\mathrm{Image});U\bigr)
> \+\\alpha_3 I(h_\psi(\mathrm{Image});Y\mid S\bigr),
> \end{equation}
> where $\alpha_1,\alpha_2, \alpha_3 >0$.  Then any global minimizer $\psi^*$ yields
>
> \begin{equation}
> Z^* = h_\psi(\mathrm{Image})
> \end{equation}
> which is identifiable in accordance with the definition of Vsual IV we defined.
>
> **Proof idea:** We further prove from the perspectives of existence and uniqueness that under these mutual information constraints, the learned visual IV is identifiable in the function space and guaranteed to be unique under invertible transformations.
>
> In addition to the theoretical support provided by the above theorems, we will also include a theoretical analysis of generalization performance in the revised version.
>
> **Q5: Experiments and analysis of the VIV-DG 's efficiency.**
>
> **R5:** Under the experimental settings described in the appendix, we supplement the analysis and experiments to evaluate the efficiency of VIV-DG, including convergence behavior, computational complexity, and GPU memory usage.
>
> - **Convergence Behavior Experiment:** We record the convergence behavior during training and plot the corresponding convergence curves.  The results show that VIV-DG converges quickly in the initial stage of causal representation learning. In the fourth stage (bias correction), the model consistently improves upon the accuracy achieved in the first stage. Additionally, in the second and third stages, the loss decreases rapidly, indicating stable and efficient training across all phases.
>
> - **Computational Complexity:** The overall time complexity of our training framework is: $T = \mathcal{O}(N \cdot E \cdot P_{\max})$ where $N$ denotes the number of training samples, $E$ is the total number of training epochs, and $P_{\max}$ represents the maximum computational cost among the modules trained in each stage. This complexity does not involve exponential or polynomial growth, making it computationally tractable.
>
> - **GPU Memory Usage:** Using ResNet-50 as the backbone, single-GPU training on an A800 GPU requires approximately 40+ GB of memory and runs stably without memory overflow or interruption.
>
> **The primary computational cost arises from the automatic visual IV learning module. However, we consider this a reasonable and acceptable trade-off, given the improvements in automation and generalization capability that it brings.**

---

### Official Review · Reviewer_ts1f · 2025-07-02

**Clarity:** 3
**Significance:** 2
**Originality:** 2
**Rating:** 4
**Confidence:** 3

**Summary:**

The paper proposes a novel framework named VIV-DG, which can learn visual instrumental variables in causal domain generalization. It identifies key visual attributes and instrumental variables, in order to retain useful non-causal information and ensures the retention through three different learning paradigms. Results showcase improved performance on causality-driven domain generalization methods and on other baselines.

**Questions:**

* Could the authors explain why the results vary in their experiments with both versions of the model and on how many iterations or different seeds the results were evaluated?
* Are there any ablations done on why the Lite version of the model performs as good as the normal model on certain domains?
* How much VRAM does it take to train this framework and how could it scale on larger backbones for domain generalization training? Are the training resources worth the increase in performance or model explanation?

**Ethical Concerns:**

["NO or VERY MINOR ethics concerns only"]

**Final Justification:**

Authors have sufficiently answered my questions. Provided they include the new experiments on the manuscript and provide the explanations, I am raising my score to 4. I still think the method is a bit heavy on the compute reesources, but it has merit for causal domain generalization.

**Limitations:**

Yes

**Quality:**

3

**Strengths And Weaknesses:**

Strengths:
* The identification of instrumental variables is novel and interesting, as encoding critical information in a way that is not detrimental towards the generalization of the model seems to provide more information.
* The pipeline descriptions is sound, providing clear explanations on how the framework can identify causal variables and non-causal variables.

Weaknesses:
* The results on the task seem a bit random. Authors cite that in PACS that the framework surpasses all baselines, when in actuality it is equal or lower in three out of four domains, despite being higher on average.
* No explanation is given on why the Lite version of the model is performing better on certain domains. There is also no experiment with the normal version of the model on VLCS.
* The framework is said to increase computational costs with larger backbones, but the performance gain on the reported datasets seems marginal for the amount of training resources.

---

> ### Author Rebuttal · Authors · 2025-07-31
>
> We sincerely appreciate your support of the novelty of our work and your valuable comments. Your suggestions help strengthen our paper. Here, we clarify and address these issues accordingly.
>
> **Questions:**
>
> **Q1: Could the authors explain why the results vary in their experiments with both versions of the model and on how many iterations or different seeds the results were evaluated?**
>
> **R1: (1) Analysis of result variations between the two versions:** The performance differences between the two model versions mainly arise from their different strategies for predicting causal factors from Visual IVs.
>
> **VIV-DG-Lite** approximates the causal factors by combining the instrumental variables with the initial causal features, which retains more original discriminative information but is more affected by confounders. Therefore, it performs better in domains with smaller distribution shifts, such as *Product* in Office-Home and *Cartoon* in PACS.
>
> In contrast, **VIV-DG** predicts causal factors via a trained regressor for bias correction, which reduces confounding effects and thus performs better in domains with larger distribution shifts, such as *Sketch* in PACS.
> **Overall, VIV-DG is more robust than VIV-DG-Lite.**
>
> **(2) Explanation regarding how many iterations or different seeds the results were evaluated.** We provide a brief report of the training epochs for VIV-DG and VIV-DG-Lite in Tables 4 and 5 of the appendix. Here, we offer a more detailed explanation for clarity: VIV-DG is trained for a total of 100 epochs, while VIV-DG-Lite is trained for 70 epochs in total, as it removes the regression stage compared to VIV-DG. Specifically, for fair comparison, we fix the number of training epochs of the main encoder (initial training + bias correction fine-tuning) to 50 for both models, which is consistent with the baseline methods. All reported results are averaged over 3 runs with different random seeds. We will clarify these settings in the revised version.
>
> **Q2: Are there any ablations done on why the Lite version of the model performs as good as the normal model on certain domains?**
>
> **R2:** We supplement ablation studies on the three mutual information loss terms (**relevance**, **independence**, and **exclusion**) for both versions of our model, VIV-DG and VIV-DG-Lite, on the PACS dataset, as shown below.
>
> #### **Tab. 1 Ablation study of VIV-DG on PACS (ResNet18)**
> |Settings| A|C|P|S| Avg.|
> |-|-|-|-|-|-|
> |w/o I(Z;S)|85.5|80.4|95.8|83.6|86.3|
> |w/o I(Z;U)|84.9|79.7|95.9|83.0|85.9|
> |w/o I(Z;Y\|S) |84.6|79.1|95.3|82.4|85.4|
> |VIV-DG |86.6|81.5|96.9|83.9|87.2|
>
> #### **Tab. 2 Ablation study of VIV-DG-Lite on PACS (ResNet18)**
> |Settings| A|C|P|S| Avg.|
> |-|-|-|-|-|-|
> |w/o I(Z;S)|85.6|79.7|95.4|82.9|85.9|
> |w/o I(Z;U)|84.7|78.4|95.5|82.4|85.3|
> |w/o I(Z;Y\|S) |84.5|78.2|95.5|81.9|85.0|
> |VIV-DG-Lite|86.1|81.8|96.6|83.3|87.0|
>
> The results show that the **exclusion** loss is the most critical, as it effectively prevents the introduction of extra confounders as instrumental variables. Meanwhile, the **relevance** and **independence** losses work together to reduce confounding effects and enhance the expressiveness of causal representations. These three loss terms collaboratively contribute to the strong performance of both the full and Lite versions. These results, together with those from other datasets, will be included in the revised version.
>
> The Lite version shares the same mutual information constraint components as the full version. **Notably, the Lite version derives new causal factors by aggregating the instrumental variables with the initial causal representations, which retains more of the original discriminative information and enables it to achieve comparable performance to the full version in certain domains.**
>
> **Q3: VRAM usage, Scalability, Training cost vs. performance.**
>
> **R3: (1) It takes how much VRAM:** By using ResNet-50 as the backbone, single-GPU training on an A800 GPU requires about 40+ GB of memory and runs stably.
>
> **(2) Scalability to larger backbones:** We provide theoretical analyses of the identifiability and learnability of visual IVs (as shown below), showing that any extracted features that satisfy the IV conditions can serve as approximate visual IVs. **This theoretical grounding ensures that our method remains valid regardless of the backbone architecture. As a result, VIV-DG scales effectively to larger backbones in domain generalization training while still producing reliable visual IVs.**
>
> - **lemma[Learnability of Visual IVs]**
> Let $h_\psi:\mathrm{Image}\to Z$ be a parameterized mapping (e.g., via a deep network), with the following training objective
> \begin{equation}
> \mathcal{L}(\psi)
> \=\
> -\alpha_1 I(Z;S\bigr)
> \+\\alpha_2 I(Z;U\bigr)
> \+\\alpha_3 I(Z;Y\mid S\bigr),
> \end{equation}
> where $\alpha_1,\alpha_2, \alpha_3 >0$.  Then any global minimizer $\psi^*$ satisfies
>
> $I(h_{\psi^*}(\mathrm{Image});S)>0$,
>
> $I(h_{\psi^*}(\mathrm{Image});U)>0$,
>
> $I(h_{\psi^*}(\mathrm{Image});\,Y\mid S)>0$,
> i.e. it exactly recovers a Visual IVs.
>
> **Proof idea:** Since $-I(Z;S)\ge -\sup_\psi I(Z;S)$,  $I(Z;U)\ge0$,  $I(Z;Y\mid S)\ge0$, any global minimizer $\psi^*$ must simultaneously (i) maximize $I(Z;S)$, (ii) drive $I(Z;U)$ to zero, and (iii) drive $I(Z;Y\mid S)$ to zero; otherwise one could reduce $\mathcal{L}$ further. By the zero‐mutual‐information conditions we get $Z\perp\ U$ and $Z\perp\ Y\mid S$, and by the maximization $I(Z;S)>0$. These three facts are exactly the three requirements of the definition of Visual IVs.
>
> - **Theorem (Identifiability for Visual IVs):** Let $(S, U, Y)$ be three random variables, where
> $S$ denotes the causal factor affecting $Y$,
> $U$ is the confounder,
> and $Y$ is the downstream label. Assume that $\mathcal H_{Z}=\{h_\psi:\mathrm{Image}\to Z\}$  is a sufficiently expressive family of mappings (e.g., one that contains all smooth bijections on a latent subspace).
> The optimize the objective
> \begin{equation}
> \mathcal{L}(\psi)
> \=\
> -\alpha_1 I(h_\psi(\mathrm{Image});S\bigr)
> \+\\alpha_2 I(h_\psi(\mathrm{Image});U\bigr)
> \+\\alpha_3 I(h_\psi(\mathrm{Image});Y\mid S\bigr),
> \end{equation}
> where $\alpha_1,\alpha_2, \alpha_3 >0$.  Then any global minimizer $\psi^*$ yields
>
> \begin{equation}
> Z^* = h_\psi(\mathrm{Image})
> \end{equation}
> which is identifiable in accordance with the definition of Vsual IV we defined.
>
> **Proof idea:** We further prove that, under these constraints, the learned visual IV is identifiable in the function space, with uniqueness guaranteed up to invertible transformations.
>
>
> **(3) Trade-off between training cost, interpretability, and automation:**
>
> **We consider the computational cost of VIV-DG to be commensurate with its higher degree of automation, enhanced interpretability, and improved scalability.** The main computational cost comes from the automatic Visual IV learning module. Although this introduces additional training cost, it significantly reduces the manual effort required to design instrumental variables and avoids reliance on domain knowledge or expert heuristics.
>
> Importantly, we supplement our work with theoretical analysis proving that the automatically learned Visual IVs strictly satisfy the IV conditions, which enhances the **interpretability** of the model.
>
> While the performance improvements over some baseline methods are moderate, our approach consistently outperforms predefined instrumental variable methods such as IV-DG in terms of accuracy and practical applicability. **Therefore, we consider the computational cost a reasonable trade-off.**
>
> Moreover, the computational cost is also a limitation we mention in the paper. To address this, we provide a simplified version, VIV-DG-Lite, which removes one training stage, significantly reducing the computational cost while still maintaining competitive accuracy.
>
> **Weaknesses:**
>
> **W1: The results seem a bit random. Authors cite that in PACS that the framework surpasses all baselines, when in actuality it is equal or lower in three out of four domains.**
>
> **R1: (1) Explanation of Experimental Results:** At first glance, the experimental results may seem less favorable on individual domains. However, it is important to note that our two versions, VIV-DG and VIV-DG-Lite, are also compared against each other. As shown in Table 1 of the paper, VIV-DG achieves the best performance only on the *Photo* domain in the PACS dataset, which may suggest limited advantage at first glance. However, **a closer examination reveals that VIV-DG is mainly outperformed by its simplified version, VIV-DG-Lite, in the domains where it ranks second.**
>
> If VIV-DG-Lite is excluded from the comparison, VIV-DG achieves the best average performance on PACS, ranks first in two domains, and second in the remaining two.
>
> Furthermore, beyond PACS, **VIV-DG achieves the best results on two domains of Digits-DG with a notable improvement in average performance, and outperforms all baselines on three out of four domains in Office-Home.**
>
> **W2: Lack of explanation for why the Lite version outperforms in some domains, and missing results of the normal version on VLCS.**
>
> **R2: (1)  Analysis of result variations between the two versions:** The performance differences and detailed analysis between VIV-DG-Lite and VIV-DG across domains are provided in our response to Question 1.
>
> **(2) Results of normal version VIV-DG on VLCS:** We thank the reviewer for pointing this out. This was our oversight. **We supplement the results of VIV-DG on the VLCS dataset**, as shown below, further demonstrating the effectiveness of VIV-DG. We will incorporate these results into Table 2 in the revised version.
> #### **Tab. 3 Results of VIV-DG on VLCS**
>
> |Methods|C|L|S|P|Avg.|
> |-|-|-|-|-|-|
> |VIV-DG-Lite|**99.2**|67.4|74.4|**77.6**| **79.7** |
> |**VIV-DG**|99.0 |**67.7**|**74.6** |77.5|**79.7** |
>
> **W3: Marginal gains vs. Computational cost.**
>
> **R3:** We provide a detailed explanation; please refer to our response to Question 3.

---

> > ### Comment · Reviewer_ts1f · 2025-08-02
> > **Response to rebuttal**
> >
> > Thank you to the authors for the response. Most of my concerns have been resolved, however I believe the compute resources are somewhat heavy for the task, given that other methodologies can run on smaller GPUs. Nevertheless, I am increasing my score to 4, given the new experiments showcased.

---

> > > ### Author Response · Authors · 2025-08-02
> > >
> > > We sincerely appreciate the reviewer’s constructive feedback and the increased score. We are pleased that the additional experiments helped address most of the concerns raised. As suggested, we will incorporate these discussions into the revised manuscript to further improve the clarity and quality of our work. Thank you once again for your valuable insights.

---

### Official Review · Reviewer_Z3TL · 2025-07-03

**Clarity:** 3
**Significance:** 3
**Originality:** 3
**Rating:** 5
**Confidence:** 3

**Summary:**

This paper introduces VIV-DG, a novel method for confounding-resistant domain generalization in image classification. The central contribution is the concept of a "Visual Instrumental Variable" (Visual IV), which proposes to automatically learn representations from image data that satisfy the conditions of an instrumental variable (IV). This method refines the causal model by partitioning non-causal factors into confounders and useful IVs. The framework consists of three main components trained via a multi-stage alternating optimization: 1) a causal feature extractor, 2) a visual IV learner that is explicitly optimized to enforce the IV conditions of relevance, independence, and exclusion using mutual information objectives, and 3) a regression predictor to generate debiased causal features. The authors claim this approach effectively mitigates the effects of unobserved confounders and outperforms existing state-of-the-art methods.

**Questions:**

1. The paper uses proxy losses to enforce the IV conditions but only evaluates success via downstream accuracy. Can you provide a more direct verification that the learned representation Z actually satisfies the IV conditions post-training? For example, please report the estimated values of I(Z;S), I(Z;U), and I(Y;Z|S) on a held-out set and compare them to a baseline without these explicit constraints. A satisfactory answer here would provide direct evidence for the central claim of the paper.
2. The Grad-CAM visualizations (Fig. 6) are confusing because I do not observe a noticeable difference between baseline and the author's method. It also does not provide definitive proof that the model is learning more "causal" representations. Can you provide a toy example showing the effectiveness of the proposed method on learning causal representations?

**Ethical Concerns:**

["NO or VERY MINOR ethics concerns only"]

**Final Justification:**

I thank the authors for the detailed additional experiments and theoretical analysis. Most of my earlier concerns have been satisfactorily addressed.

The only remaining major concern is the dependence on handcrafted masking: Although the δ sensitivity experiment on Digits-DG shows reasonable robustness, the grayscale-difference threshold mask remains a core dependency, which partially conflicts with the “automatic” positioning of the IV learning process. I suggest highlighting this limitation more prominently in the final version and prioritizing fully automated mask generation in future work.

Overall, I find that the rebuttal addresses most issues, so I would raise my score to accept.

**Limitations:**

The authors acknowledge the computational cost of the regressor (Remark 2) and briefly mention generic failure modes (Appendix H).

**Paper Formatting Concerns:**

N/A.

**Quality:**

2

**Strengths And Weaknesses:**

### **Strengths**:

1. The core idea of *learning* an instrumental variable representation directly from visual data, rather than relying on predefined, potentially invalid instruments (as in IV-DG), is a significant and novel conceptual contribution to the causal DG literature. The proposed structural causal model in Figure 1c, which explicitly separates non-causal factors into confounders and IVs, provides a more nuanced and promising theoretical framing for the problem than the standard approach of discarding all non-causal information.

2. The translation of the three classical IV conditions into differentiable loss functions using mutual information neural estimation and KL-divergence is technically sound and well-motivated. The paper is also thorough in its experimental evaluation, testing the proposed method across four standard DG benchmarks (Digits-DG, PACS, Office-Home, VLCS) and comparing it against a wide range of baselines.

### **Weaknesses**

1. The "automatic" IV learner depends critically on a heuristic masking procedure to separate IV-relevant features (texture/color) from causal features (shape). This mask is generated by thresholding the pixel-wise difference between grayscale images (Appendix D). This is a brittle, handcrafted step at the core of an otherwise "automatic" learning pipeline. There is no analysis of its robustness (e.g., to the choice of threshold δ) or verification that it truly isolates the intended features.

2. The primary motivation for this work is to improve upon methods like IV-DG that use predefined instruments. However, the comparison to IV-DG is limited, and the authors fail to include it on the Digits-DG benchmark. A more direct and analytical comparison showing why the learned IV succeeds where the predefined one fails is needed, rather than just reporting aggregate scores.

3. The reported performance gains over the strongest baselines are marginal on several key datasets. For instance, the improvement on PACS is approximately 0.4% over FAGT, and on Office-Home, it is 0.3% over CIRL. While SOTA is achieved, these modest gains, coupled with the model's immense complexity and questionable assumptions, fail to provide a compelling conceptual takeaway. The Grad-CAM visualizations (Fig. 6) are confusing because I do not observe a noticeable difference between baseline and the author's method. It also does not provide definitive proof that the model is learning more "causal" representations.

---

> ### Author Rebuttal · Authors · 2025-07-31
>
> We sincerely appreciate your support of the novelty of our work and your valuable comments. The suggestions help improve our work, and we clarify and address the raised concerns as follows.
>
> **Questions:**
>
> **Q1: Direct verification that the learned IV representation satisfies the IV conditions.**
>
> **R1: We provide validation from both experimental and theoretical perspectives:**
>
> **(1) Experiments on the estimated values of mutual information:** We conduct additional experiments to estimate and quantify the mutual information terms $I(Z;S)$, $I(Z;U)$ and $I(Y;Z|S)$. The results indicate that the learned Visual IV satisfies the relevance, independence, and exclusion conditions, confirming its validity.
> #### **Tab. 1 Estimated values of mutual information on PACS**
> ||A|C|
> |-|-|-|
> |I(Z;S)|0.86|0.84|
> |I(Z;U)|0.21|0.25|
> |I(Z;Y\|S)|0.25|0.27|
>
> **(2) Ablation studies on the three mutual information loss:** We supplement ablation studies on the three mutual information loss terms (**relevance**, **independence**, and **exclusion**) for both versions of our model on the PACS dataset, as shown below. The results show that the **exclusion** loss is the most critical, as it effectively prevents the introduction of extra confounders as IVs. Meanwhile, the **relevance** and **independence** losses help reduce confounding effects and enhance the expressiveness of causal representations. These results, together with those from other datasets, will be included in the revised version.
> #### **Tab. 2 Ablation study of VIV-DG on PACS (ResNet18)**
> |Settings|A|C|P|S| Avg.|
> |-|-|-|-|-|-|
> |w/o I(Z;S)|85.5|80.4|95.8|83.6|86.3|
> |w/o I(Z;U)|84.9|79.7|95.9|83.0|85.9|
> |w/o I(Z;Y\|S) |84.6|79.1|95.3|82.4|85.4|
> |VIV-DG |86.6|81.5|96.9|83.9|87.2|
>
> #### **Tab. 3 Ablation study of VIV-DG-Lite on PACS (ResNet18)**
> |Settings|A|C|P|S| Avg.|
> |-|-|-|-|-|-|
> |w/o I(Z;S)|85.6|79.7|95.4|82.9|85.9|
> |w/o I(Z;U)|84.7|78.4|95.5|82.4|85.3|
> |w/o I(Z;Y\|S) |84.5|78.2|95.5|81.9|85.0|
> |VIV-DG-Lite|86.1|81.8|96.6|83.3|87.0|
>
> **(3) Theoretical Support:**
>
> We supplement a theoretical analysis to support the learnability and identifiability of Visual IVs. Below we summarize the main theoretical results and proof ideas; full details will be included in the revised version:
>
> - **Lemma  (Learnability of Visual IVs):** Let $h_\psi:\mathrm{Image}\to Z$ be a parameterized mapping (e.g., via a deep network), with the following training objective
> \begin{equation}
> \mathcal{L}(\psi)
> \=\
> -\alpha_1 I(Z;S\bigr)
> \+\\alpha_2 I(Z;U\bigr)
> \+\\alpha_3 I(Z;Y\mid S\bigr),
> \end{equation}
> where $\alpha_1,\alpha_2, \alpha_3 >0$.  Then any global minimizer $\psi^*$ satisfies
>
> $I(h_{\psi^*}(\mathrm{Image});S)>0$,
>
> $I(h_{\psi^*}(\mathrm{Image});U)>0$,
>
> $I(h_{\psi^*}(\mathrm{Image});\,Y\mid S)>0$,
> i.e. it exactly recovers a Visual IV.
>
> **Proof idea:** We define an objective function that combines the three mutual information terms, and prove that any global minimizer must satisfy all three conditions, thereby yielding a valid Visual IV.
>
> - **Theorem (Identifiability for Visual IVs):** Let $(S, U, Y)$ be three random variables, where
> $S$ denotes the causal factor affecting $Y$,
> $U$ is the confounder,
> and $Y$ is the downstream label. Assume that $\mathcal H_{Z}=\{h_\psi:\mathrm{Image}\to Z\}$  is a sufficiently expressive family of mappings (e.g., one that contains all smooth bijections on a latent subspace).
> The optimize the objective
> \begin{equation}
> \mathcal{L}(\psi)
> \=\
> -\alpha_1 I(h_\psi(\mathrm{Image});S\bigr)
> \+\\alpha_2 I(h_\psi(\mathrm{Image});U\bigr)
> \+\\alpha_3 I(h_\psi(\mathrm{Image});Y\mid S\bigr),
> \end{equation}
> where $\alpha_1,\alpha_2, \alpha_3 >0$.  Then any global minimizer $\psi^*$ yields
>
> \begin{equation}
> Z^* = h_\psi(\mathrm{Image})
> \end{equation}
> which is identifiable in accordance with the definition of Vsual IV we defined.
>
> **Proof idea:** We further prove that, under these constraints, the learned visual IV is identifiable in the function space, with uniqueness guaranteed up to invertible transformations.
>
> **Q2: Explanation of Visualizations and Toy Example.**
>
> **R2: (1) Explanation for the Grad-CAM visualizations:** We apologize for the unclear explanation of the visualization experiments, and we clarify the details as follows. We compare our VIV-DG with CIRL, a strong causal domain generalization baseline, in the visualization experiments. We intentionally select the *Art Painting* domain—where the classification accuracies of the two methods are similar—to demonstrate that our method better captures causal regions. Even if the improvement is not visually striking, it still shows progress in identifying causal cues.
>
> To further demonstrate the causal representation learning ability of VIV-DG, we extend the Grad-CAM visualizations to the *Cartoon* and *Sketch* domains. (We cannot display the images here.) The results further show that VIV-DG focuses more on the central structure and semantically relevant object regions, which is consistent with the superior performance of VIV-DG over CIRL in these domains.
>
> **Additional visualization evidence:** To verify that the learned IVs alleviate confounding effects, we visualize the Grad-CAM maps of causal features before and after IV-based correction. The results show that the model captures more robust and semantically meaningful regions after using the visual IVs.
>
> **(2) Toy example:** Consider an image of a dog on the grass, where the dog represents the causal feature $S$ and the grass serves as the confounder $U$. Methods like CIRL aim to disentangle causal features but do not explicitly remove confounding bias, so the learned representations may still retain spurious signals (e.g., grass color).
> In contrast, our method learns a visual IV $Z$ that satisfies the IV conditions and uses it to predict a new causal feature $\hat{S}$ via a regressor, which is unbiased by confounders. Since $Z$ is explicitly independent of $U$ and influences $Y$ only through $S$, the predicted $\hat{S}$ is free from confounding interference. It is determined solely by $Z$ and guided by supervision from $Y$, which encourages $\hat{S}$ to capture mainly causal information. This results in more robust and accurate causal representations.
>
> **Weaknesses:**
>
> **W1: Robustness analysis of the masking process.**
>
> **R1:** We add a sensitivity experiment on the Digits-DG dataset using threshold  δ ranging from 20 to 40, as shown below, and observe minimal performance variation, indicating good robustness.
> #### **Tab. 4 Sensitivity to δ on Digits-DG**
> |δ|MN|MM|SVHN|SYN| Avg.|
> |-|-|-|-|-|-|
> |20|97.2|66.5|76.9|90.4|82.8|
> |25|97.0|66.7|76.3|90.8|82.7|
> |30|97.6|67.1|77.4|91.6|83.4|
> |35|97.4|67.3|77.0|91.8|83.4|
> |40|97.0|66.5|76.6|90.6|82.7|
>
> We use a grayscale difference δ of 30 to generate the mask, which is a common empirical value in image processing. **This simple strategy helps reduce computational cost while maintaining efficiency.** In future work, we plan to develop an automated mask generation mechanism to further enhance robustness and automation.
>
> **W2: Lack of comparison with IV-DG on the Digits-DG dataset and analysis of their respective advantages and disadvantages.**
>
> **R2: (1) Explanation for not including a comparison with IV-DG on Digits-DG:** Following common practice, we report IV-DG results directly from the original paper for fair comparison. IV-DG is only evaluated on PACS and Office-Home in its original paper, and not on other benchmarks such as Digits-DG. Therefore, we do not include the results of IV-DG on the Digits-DG benchmark. Notably, compared to IV-DG, our method achieves improvements of +2.6% on PACS and +3.0% on Office-Home, demonstrating a clear advantage.
>
> **(2) Learnable IVs vs. Predefined IVs:** Regarding the limitations of predefined IVs: Classical IV assumptions—especially the exclusion condition—are strict and often difficult to verify in practice. For instance, IV-DG selects data from one domain as the IV for another, which can easily violate the exclusion condition. Consider using a cartoon image of a dog as the IV for an art-style image of the same dog. Although the domains differ, the semantic label is the same, and features extracted from the cartoon image may directly predict the label "dog", thus violating the exclusion requirement and leading to weak or invalid IVs.
> In contrast, our learnable IV approach explicitly enforces the IV conditions during training, thereby avoiding such violations and yielding more reliable causal representations.
>
> **W3: The performance improvements are relatively modest, the model introduces additional complexity, and the Grad-CAM visualizations do not clearly illustrate whether more causal representations are being learned.**
>
> **R3: (1) Computational cost.** **We consider the computational cost of VIV-DG to be a reasonable trade-off for achieving both automation and scalability.** The main computational cost of VIV-DG comes from the automatic visual IV learning module. Although this introduces additional training cost, it substantially reduces the manual effort required to design IVs, which is often necessary in predefined IV-based methods. It also avoids reliance on domain knowledge or expert heuristics. Despite achieving only moderate gains over some baselines, it consistently outperforms predefined IV-based methods like IV-DG in both performance and practicality.
>
> Moreover, the computational cost is also a limitation we mention in the paper. To address this, we provide a simplified version, VIV-DG-Lite, which removes one training stage, significantly reducing the computational cost while still maintaining competitive accuracy.
>
> **(2) Theoretical Support:** We provide a theoretical analysis of the identifiability of visual IV, please see our response to Question 1. This further supports the learnability and validity of our approach.
>
> **(3) Grad-CAM visualizations:** For clarification and analysis of the Grad-CAM visualizations, please see our response to Question 2.

---

### Comment · Area_Chair_XNn3 · 2025-08-07

Dear Reviewers,

Thank you for your efforts in reviewing the submission.

The authors have submitted their rebuttals in response to your comments. Please ensure that you engage with the rebuttal and provide a response before selecting “Mandatory Acknowledgement.”

We have noticed that some reviewers have submitted a “Mandatory Acknowledgement” without posting any discussion or feedback to the authors. Please note that this is not permitted.

Please note “Mandatory Acknowledgement” button is to be submitted only when reviewers fulfill all conditions below (conditions in the acknowledgment form):
1. read the author rebuttal
2. engage in discussions (reviewers must talk to authors, and optionally to other reviewers and AC - ask questions, listen to answers, and respond to authors)
3. fill in "Final Justification" text box and update “Rating” accordingly (this can be done upon convergence - reviewer must communicate with authors first)

---

### Note · Authors · 2025-08-12

Dear Area Chairs and Reviewers,

We sincerely thank you for the comprehensive review of our work and the valuable feedback provided. We are pleased to note that three reviewers (Z3TL, ts1f, and ubcT) unanimously recognized the **novelty of our approach**, with ubcT explicitly commending our **clear research motivation** as well as the **clear structure and concepts** of the paper. We also note reviewer z3cX's acknowledgment of the **theoretical interest** in our work. These positive comments affirm the core contributions of our work and indicate that our idea of automatically learning visual instrumental variables to enhance domain generalization has been positively received by the reviewers, which greatly encourages us to further pursue this line of research.

Reviewer concerns primarily centered on additional experiments and time cost:
- **Experimental requests**: All suggested comparisons have been carefully added, consistently demonstrating our method's effectiveness and robustness.
- **Time cost**: While automatic IV learning incurs computational cost, it mitigates the substantial expert-dependence and potential bias inherent in manual IV design. We view this as a trade-off for achieving automation, scalability, and effectiveness. Moreover, the computational cost is explicitly acknowledged as a limitation in our paper. To address it, we also provide a simplified variant, VIV-DG-Lite, which significantly reduces the computational cost while maintaining competitive accuracy.

We note that reviewer z3cX questioned the accuracy of our structural causal model (SCM). We have addressed this concern in detail in our rebuttal. We believe the concern may stem from a different interpretation of certain model details, and we have reiterated the rationale behind our design along with illustrative examples to clarify. Notably, no other reviewers questioned our SCM, suggesting its general acceptance among peers.

We hope these clarifications help convey the value of our contributions. We will integrate reviewer suggestions and refine relevant sections in the revised manuscript.

We once again thank all reviewers and the committee for their time, effort, and thoughtful feedback, and we look forward to your final decision.

Sincerely,

The Authors

---

### Decision · Program_Chairs · 2025-09-17

**Decision:**

Accept (poster)

**Comment:**

The paper introduces the novel concept of Visual IVs, which is both innovative and impactful for domain generalization. The motivation is clear, and the framework is logically sound. The authors made strong efforts during rebuttal, including new experiments and theoretical clarifications. I suggest adding the DomainBed results in the final version. Overall, I recommend acceptance.